# COSDA: Counterfactual-based Susceptibility Risk Framework for Open-Set Domain Adaptation

**Wenxu Wang** [* 1]   **Rui Zhou** [* 2]   **Jing Wang** [3]   **Yun Zhou** [2]   **Cheng Zhu** [2]   **Ruichun Tang** [1]   **Bo Han** [4]
**Nevin L. Zhang** [5]

## Abstract

Open-Set Domain Adaptation (OSDA) aims to transfer knowledge from the labeled source domain to the unlabeled target domain that contains unknown categories, thus facing the challenges of domain shift and unknown category recognition. Little exploration has been conducted on causal-inspired theoretical frameworks for OSDA. To fill this gap, we introduce the concept of *Susceptibility* and propose a novel **C**ounterfactual-based susceptibility risk framework for **OSDA**, termed **COSDA**. Specifically, COSDA consists of three novel components: (i) a *Susceptibility Risk Estimator (SRE)* for capturing causal information, along with comprehensive derivations of the computable theoretical upper bound, forming a risk minimization framework under the OSDA paradigm; (ii) a *Contrastive Feature Alignment (CFA)* module, which is theoretically proven based on mutual information to satisfy the *Exogeneity* assumption and facilitate cross-domain feature alignment; (iii) a *Virtual Multi-unknown-categories Prototype (VMP)* pseudo-labeling strategy, providing label information by measuring how similar samples are to known and multiple virtual unknown category prototypes, thereby assisting in open-set recognition and intra-class discriminative feature learning. Extensive experiments demonstrate that our approach achieves state-of-the-art performance.

---

[*]Equal contribution  [1]College of Computer Science and Technology, Ocean University of China, Qingdao, China [2]National Key Laboratory of Information Systems Engineering, National University of Defense Technology, Changsha, China [3]Department of Mechanical Engineering, University of British Columbia, Vancouver, Canada [4]Department of Computer Science, Hong Kong Baptist University, Hong Kong, China [5]Department of Computer Science and Engineering, The Hong Kong University of Science and Technology, Hong Kong, China. Correspondence to: Ruichun Tang <tangruichun@ouc.edu.cn>.

*Proceedings of the 42nd International Conference on Machine Learning*, Vancouver, Canada. PMLR 267, 2025. Copyright 2025 by the author(s).

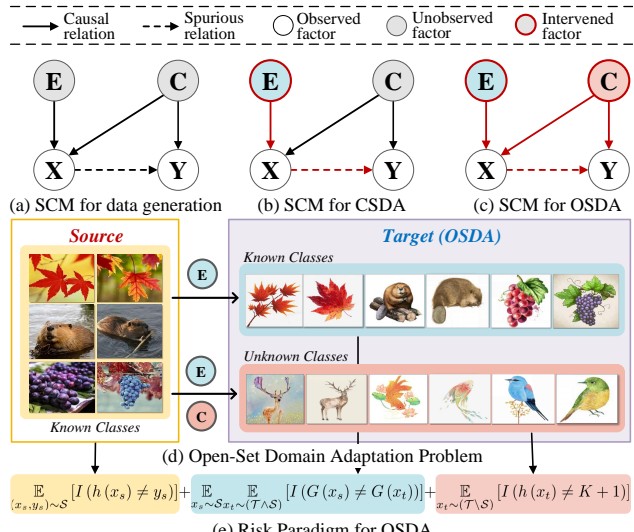

*Figure 1.* (a) Causal graph for data generation, (b) causal graph for domain shift, where red arrows describe the distribution of data $(\mathbf{X}, Y)$ is shifted because environment variables $\mathbf{E}$ are intervened in a new unseen domain, and (c) causal graph for OSDA, where red arrows indicate that the intervened environment variables $\mathbf{E}$ and intervened causal variables $\mathbf{C}$ lead to unknown classes; (d) shows how images changed with the interventions on $\mathbf{E}$ and $\mathbf{C}$; (e) the risk paradigm for OSDA, including source supervised risk, feature alignment risk, and open-set space risk.

## 1. Introduction

Traditional supervised learning algorithms heavily rely on the in-distribution assumption (Murphy, 2012), which assumes that training and test data are sampled from the same distribution. However, this assumption is not consistent in wild environments, where domain shift (Zhang et al., 2013) and category shift (Qu et al., 2023) lead to the failure of supervised learning algorithms. Considering domain shift, Close-Set Domain Adaptation (CSDA) aims to transfer a model from a labeled source domain to an unlabeled target domain under the assumption of a shared label space (Ganin & Lempitsky, 2015; Cui et al., 2024). Existing domain adaptation methods can be broadly categorized into two main streams: metric learning (Long et al., 2016; Wei et al., 2024) and adversarial learning (Ganin et al.; Li et al., 2018).

However, in test environments, encountering novel classes is common, exposing the limitations of the shared label space assumption. To address both domain shift and class shift, researchers have begun to explore a different learning setting known as Open-Set Domain Adaptation (OSDA) (Saito et al., 2018; Wang et al., 2024; Fang et al., 2020), which integrates CSDA with open-set recognition. OSDA assumes that the label space of the source domain is a subset of that of the target domain, meaning that the model is expected to encounter unseen classes during real-world applications.

A body of work has focused on OSDA, and the risk formation of these studies is illustrated in Fig. 1 (e). Here, $\mathbb{E}_{x_s \sim \mathcal{S}}[I(h(x_s) \neq y_s)]$ aims at unbiased supervised learning of source domain (Bucci et al., 2020; Li et al., 2023); $\mathbb{E}_{x_s \sim \mathcal{S}} \mathbb{E}_{x_t \sim (\mathcal{T} \wedge \mathcal{S})}[I(G(x_s) \neq G(x_t))]$ denotes feature alignment to address domain shift (Ganin et al.; Luo et al., 2020; Wei et al., 2024); and $\mathbb{E}_{x_t \sim (\mathcal{T} \setminus \mathcal{S})}[I(h(x_t) \neq K+1)]$ represents the minimization of open-set risk to achieve unknown class recognition (Saito et al., 2018), where most methods require the design of unknown class recognition strategies (Liu et al., 2019; Saito, 2021; Qu et al., 2023).

Causal models have been widely utilized to achieve domain alignment (Arjovsky et al., 2020; Lu et al., 2021; Yuan et al., 2024; Yue et al., 2021) due to the invariance of causal features. Most causal models divide latent features into environment-specific features ($\mathbf{E}$) and causal features ($\mathbf{C}$), assuming that $E \perp C$ and $E \perp Y \mid C$ (Yang et al., 2024; Lv et al., 2022), as shown in Fig. 1(b). Domain shift in causal diagrams arises from interventions on domain-specific variables, which alter their distributions and consequently affect the joint distribution of the dataset, while $P(Y \mid C)$ remains stable(Yuan et al., 2024). However, for OSDA, there has been little exploration of causal graphs and a comprehensive causal-based risk theory. This is the gap that we aim to fill.

We argue that causal features still play a crucial role in OSDA. When causal features are not intervened, the conditional probability of the label $P(Y|\mathbf{C})$ will remain stable. However, significant shifts in causal features can lead to category shifts, explaining the emergence of new categories. Therefore, the model must be sufficiently concentrated on core causal features. Based on this, we propose a promising OSDA method using essential causal information, which builds upon the probability of *Susceptibility*. First, we model an evaluator to assess and optimize the susceptibility of features. Second, to assess risk with limited data, an OSDA risk framework is provided through theoretical analysis. For the identifiability of counterfactual-based susceptibility risk, characteristics must fulfill two causal assumptions: *Exogeneity* and *Monotonicity*. To ensure that extracted features fit these causal assumptions, we propose two novel strategies: a mutual information theory-based Contrastive-

inspired Feature Alignment (CFA) optimization objective to align cross-domain features and a pseudo-labeling strategy with Virtual Multi-unknown-categories Prototypes (VMP) instead of treating the unknown class as a single category.

The main contributions of this paper are as follows:

**Promising Way**: We propose a structural causal model within the OSDA paradigm and perform theoretical derivations and algorithm design centered on the counterfactual probability of susceptibility. This addresses its optimizability, evaluability, and identifiability, thus presenting a principled causal framework for OSDA tasks.

**Systematic Theoretical Framework**: We introduce a novel counterfactual-based risk—*Susceptibility risk*—along with its evaluator. A theoretically computable upper bound for the target domain's susceptibility risk is derived within the OSDA paradigm.

**Innovative Techniques**: We identify an optimization objective that satisfies the *Exogeneity* causal assumption, which is recognized as maximizing the mutual information between the target sample and the source domain prototype. Based on this, we propose a *Virtual Multi-unknown-categories Prototype* (VMP) pseudo-labeling strategy and a *Contrastive-inspired Feature Alignment* (CFA) module.

**Comprehensive Experiments**: We validate the effectiveness of the model on three benchmark datasets, achieving improvements of 2.9%, 2.2%, and 1.0%, respectively, compared to state-of-the-art (SOTA) algorithms. Ablation studies and experiments on synthetic datasets confirm the effectiveness of each proposed module.

## 2. Preliminaries

Causal inference based on counterfactual distributions involves evaluating the effects of interventions on outcomes and seeks to identify variables whose alterations significantly influence the actual values of outcomes. To quantify this influence, we introduce the concept of *Probability of Counterfactual*, which is formally defined as below.

**Definition 1** (Probability of Counterfactual (Pearl, 2009)). *Let the invariant representation of causal variables $\mathbf{C}$ for label $y$ be $\mathbf{c}$, and $\mathbf{c}'$ is the specific implementation of $\mathbf{C}$, where $\mathbf{c} \neq \mathbf{c}'$. The probability that $Y$ changes from $Y \neq y$ to $Y = y$ when $\mathbf{C}$ is altered from $\mathbf{c}'$ to $\mathbf{c}$ is*

$$PC = Pr(Y_{do(\mathbf{C}=\mathbf{c})} = y \mid \mathbf{C} = \mathbf{c}', Y \neq y). \quad (1)$$

Two causal definitions, *Exogeneity* and *Monotonicity*, have been introduced by Pearl (Pearl, 2009) for identifiability. *Exogeneity* represents a condition of *no confounder*, ensuring the vanishing of differences between the intervention distribution and the conditional distribution. *Monotonicity* further elucidates the monotonic direction of causal effects. The definitions of *Exogeneity* and *Monotonicity* are detailed

in Appendix B.

Based on *Exogeneity* and *Monotonicity*, the identifiable distribution of Eq. (1) is described by the following lemma, which is defined as susceptibility.

**Lemma 1** (Probability of susceptibility ((Pearl, 2009))). *If* **C** *is exogenous relative to* $Y$*, and* $Y$ *is monotonic relative to* **C***, then the probability that* $Y$ *is susceptible to* **C** *is*

$$PS = \frac{Pr(Y = y | \mathbf{C} = \mathbf{c}) - Pr(Y = y | \mathbf{C} = \mathbf{c}')}{1 - Pr(Y = y | \mathbf{C} = \mathbf{c}')}. \quad (2)$$

The proof of lemma 1 is established by the logical analysis carried out by Pearl (Pearl, 2009). A characteristic with a higher $PS$ typically encompasses more significant causal information. Appendix A presents examples to illustrate where the susceptibility is high and where it is low. Furthermore, susceptibility risk, compared to traditional causal effect evaluations (Pearl, 2009), proves to be more effective in identifying core causal features. This indicates that susceptibility risk functions as a more discerning indicator for identifying causal links, hence improving the model's capacity to concentrate on critical causal information.

Additionally, causality posits the *Semantic Separability* assumption of features, meaning that features with distinct semantics maintain a certain distance and a certain feature does not correspond to multiple semantic meanings simultaneously. To meet this condition, we define the stability loss $\mathcal{L}_{\text{STA}}$ (Yang et al., 2024) as follows:

$$\mathcal{L}_{\text{STA}} = \epsilon - \| \mathbf{c} - \mathbf{c}' \|_2. \quad (3)$$

where $\epsilon$ represents the degree of intervention on the causal features. $\mathcal{L}_{\text{STA}}$ indicates that the semantic meaning is distinguishable between $\mathbf{c}$ and $\mathbf{c}'$, requiring that the degree of intervention must be sufficiently large to induce semantic changes in features, thereby avoiding inherently unstable learning.

## 3. Methodology

### 3.1. Learning Setup

Assume that we have the labeled source data $\mathcal{D}_s = \{\mathbf{X}_s, Y_s\} = \{(\mathbf{x}_{si}, y_{si})\}_{i=1}^{n_s} \sim \mathcal{S}$ and unlabeled target data $\mathcal{D}_t = \{\mathbf{X}_t\} = \{(\mathbf{x}_{ti})\}_{i=1}^{n_t} \sim \mathcal{T}_{\mathbf{X}}$, where $\mathcal{S}$ is the joint probability distribution of the source domain, $\mathcal{T}$ is the marginal distribution of the target domain, with $n_s$ and $n_t$ indicating the size of the source and target datasets, respectively. **Open-set Domain Adaptation (OSDA)** allows the source label space $\mathcal{Y}_s = \{1, ..., K\}$ and the target label space $\mathcal{Y}_t = \{\mathcal{Y}_s, K + 1\}$, where $K + 1$ denotes the unknown class. Given the i.i.d. samples drawn from $\mathcal{S}$ and $\mathcal{T}_{\mathbf{X}}$, the goal of OSDA is to train a model that can classify the samples from the known classes and recognize the samples from the unknown class correctly.

### 3.2. Susceptibility Risk Modelling

This section presents the *Susceptibility Risk Estimator (SRE)* for representation learning in the target domain. First, we introduce a concept, *Probability of Intervention Relevance (PIR)*, which is used to quantify the probability of $Y$ after **C** has been intervened.

**Definition 2** (Probability of Intervention Relevance (PIR)). *Given a test domain* $\mathcal{T}$*, if* **E** $\perp$ **C***, and* **E** $\perp Y \mid$ **C***, the probability of intervention relevance (PIR) is defined as:*

$$PIR_t(\mathbf{c}') = P_t(Y = y | \mathbf{C} = \mathbf{c}'), \quad (4)$$

*and the Probability of Observational Relevance (POR) is defined as:*

$$POR_t(\mathbf{c}) = P_t(Y = y \mid \mathbf{C} = \mathbf{c}). \quad (5)$$

Then, we define the estimation of Susceptibility risk.

**Proposition 1** (Susceptibility Risk (SR)). ***Observational relevance risk*** $O_t(\mathbf{c})$ *and the* ***intervened relevance risk*** $O_t(\mathbf{c}')$ *are defined as:*

$$O_t(\mathbf{c}) = \underset{(\mathbf{x},y)\sim\mathcal{T}}{\mathbb{E}} \underset{\mathbf{c}\sim P_t(\mathbf{C}|\mathbf{x})}{\mathbb{E}} P(Y \neq y \mid \mathbf{c}), \quad (6)$$

$$O_t(\mathbf{c}') = \underset{(\mathbf{x},y)\sim\mathcal{T}}{\mathbb{E}} \underset{\mathbf{c}'\sim P_t(\mathbf{C}'|\mathbf{x})}{\mathbb{E}} P(Y \neq y \mid \mathbf{c}'). \quad (7)$$

*Then, susceptibility risk* $\text{SR}_t$ *based on Lemma 1 is formally defined and we derived a new formulation of* $\text{SR}_t$ *as:*

$$\text{SR}_t(\mathbf{c}, \mathbf{c}') = 1 - PS := O_t(\mathbf{c}) - O_t(\mathbf{c}'). \quad (8)$$

The induction is detailed in Appendix B.1.

### 3.3. OSDA Risk Framework with Susceptibility

In this section, we introduced several theorems to gain an evaluable upper bound for $\text{SR}_t$.

**Linking open-set recognition risks and closed-set classification risks.**

Assume supp($\mathcal{S}$) is the shared support set; the target domain in OSDA has a subsample set $\{\mathbf{X}, Y\} \notin$ supp($\mathcal{S}$). Therefore, $\text{SR}_t$ can be rewritten as:

**Theorem 1.** *Given the source domain distribution* $\mathcal{S}$ *and the target domain distribution* $\mathcal{T}_{\mathbf{X}}$*, the susceptibility risk on the test domain* $\text{SR}_t(\mathbf{c}, \mathbf{c}')$ *can be expressed as:*

$$\text{SR}_{\mathcal{T}}(\mathbf{c}, \mathbf{c}') = \underset{(\mathbf{x},y)\sim\mathcal{T}}{\mathbb{E}} [I(\mathbf{x},y) \in supp(\mathcal{S})] \Bigg[ \underset{\mathbf{c}\sim P_t(\mathbf{C}|\mathbf{X}=\mathbf{x})}{\mathbb{E}}$$
$$P(Y \neq y \mid \mathbf{C} = \mathbf{c}) - \underset{\mathbf{c}'\sim P_t(\mathbf{C}'|\mathbf{X}=\mathbf{x})}{\mathbb{E}}$$
$$P(Y \neq y \mid \mathbf{C} = \mathbf{c}') \Bigg] + \mathbb{E}_{(\mathbf{x},y)\sim\mathcal{T}} [I(\mathbf{x},y) \notin supp(\mathcal{S})]$$
$$\Bigg[ \underset{\mathbf{c}\sim P_t(\mathbf{C}|\mathbf{X}=\mathbf{x})}{\mathbb{E}} P(Y \neq K+1 \mid \mathbf{C} = \mathbf{c}) -$$
$$\underset{\mathbf{c}'\sim P_t(\mathbf{C}'|\mathbf{X}=\mathbf{x})}{\mathbb{E}} P(Y \neq K+1 \mid \mathbf{C}' = \mathbf{c}') \Bigg]$$
$$:= \pi_{t \wedge s}(\mathbf{X}, Y) + \pi_{t \setminus s}(\mathbf{X}, Y), \quad (9)$$

where supp($\mathcal{S}$) is the support set of the source domain distribution. $P_t(\mathbf{X}, Y \notin$ supp($\mathcal{S}$)) shares the same idea as the Open Space Risk (Scheirer et al., 2012), quantifying how

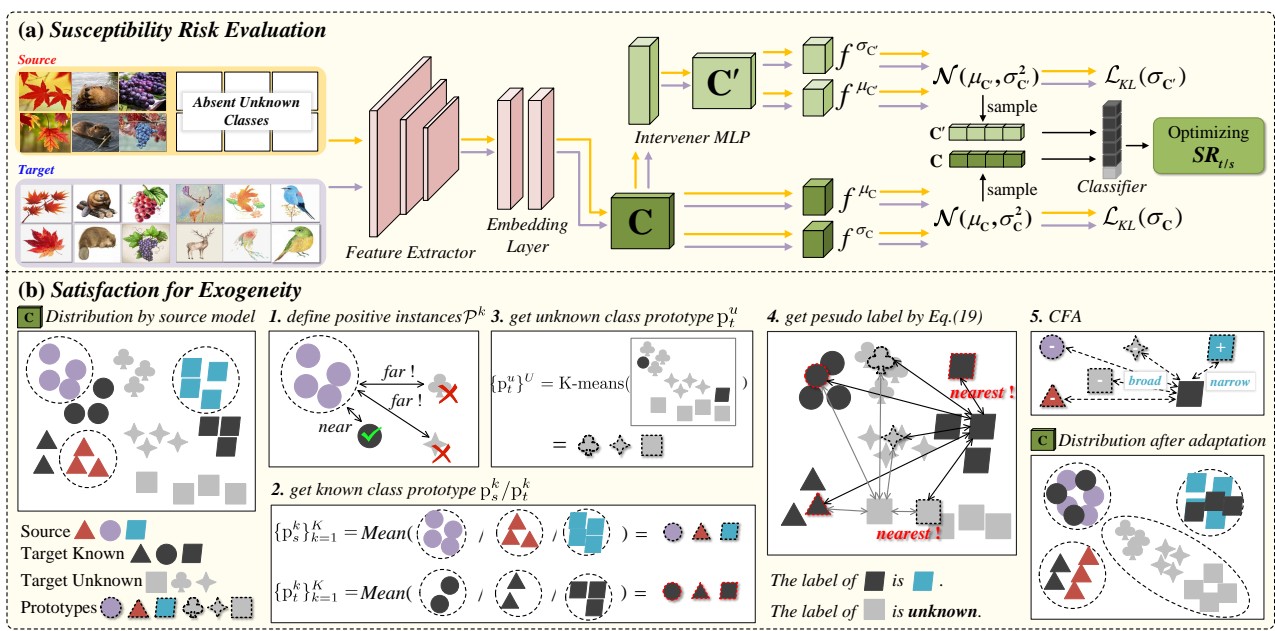

*Figure 2.* An overview of our proposed COSDA. (a) describes the deployment of feature interventions. We add an intervention module utilizing Multi-Layer Perception (MLP) following feature $C$, executing an intervention via a nonlinear transformation of feature $C$. The feature distribution is a Gaussian distribution with parameterized mean and variance, as popularized by VIB (Kingma et al., 2015). (b) The VMP and CFA are proposed to address *Exogeneity*. The central idea of *VMP* (1.-4.) is to build the centroids of all known and unknown classes, and then generate the pseudo label for samples by comparing the distance between samples and the centroids. *CFA* (5.) introduces the concept of feature alignment, which reduces the distance to the appropriate class centroid while maximizing the distance to the centroids of other classes, establishing class-level cross-domain feature alignment. Through adaptation, we align the features of known classes while also learning the decision boundary for unknown classes.

open the OSDA problem setting is. $\pi_{t\setminus s}(\mathbf{X}, Y)$ accumulates the open-set recognition error.

**Linking Target Risks and Source Risks.**

In the OSDA task, only the labeled observations of the source domain $\mathcal{S}$ and the unlabeled observations of the target domain $\mathcal{T_X}$ are provided. Consequently, direct assessment of susceptibility risk in the target domain is unattainable. To resolve this issue, we introduce the expected disagreement $d_{\mathcal{T_x}}$ and the expected joint error $e_{\mathcal{S}}$ (Lacasse et al., 2006), which needs no label information of the target domain. Then, we propose Theorem 2 using $\beta$-divergence (Germain et al., 2016) and Hölder's Inequality. Proof for Theorem 2 is provided in Appendix B.3.

**Theorem 2.** *Given the source domain distribution $\mathcal{S}$ and the target domain distribution $\mathcal{T_X}$, an invariant representation inference model $G$, an intervention model $G'$, and a classifier h, the susceptibility risk on the test domain $\mathrm{SR}_t(\mathbf{c}, \mathbf{c}')$ is upper bounded by:*

$$\mathrm{SR}_{\mathcal{T}}(\mathbf{c}, \mathbf{c}') < \frac{1}{2}(d_{\mathcal{T_x}}(\mathbf{c}) - d_{\mathcal{T_x}}(\mathbf{c}'))$$
$$+ \lim_{q \to \infty} \beta_q(\mathcal{T}\|\mathcal{S})(e_{\mathcal{S}}(\mathbf{c}) - e_{\mathcal{S}}(\mathbf{c}')) \quad (10)$$
$$+ \pi_{t\setminus s}(\mathbf{X}, Y),$$

*where*

$$\mathcal{L}_t := d_{\mathcal{T_X}}(\mathbf{c}) - d_{\mathcal{T_X}}(\mathbf{c}') + \pi_{t\setminus s}(\mathbf{X}, Y), \quad (11)$$
$$\mathcal{L}_s := e_{\mathcal{S}}(\mathbf{c}) - e_{\mathcal{S}}(\mathbf{c}'). \quad (12)$$

**Linking Empirical Risks and Expected Risks.** The expected risk $\mathrm{SR}(\mathbf{c}, \mathbf{c}')$ cannot be calculated since the distributions $\mathcal{S}$ and $\mathcal{T_X}$ are not provided. We define the estimation distributions on the available data $\mathcal{D}$ as $\bar{P}(Y = y \mid \mathbf{C} = \mathbf{c})$, $\bar{P}(Y = y \mid \mathbf{C}' = \mathbf{c}')$, $\bar{P}(Y = y \mid \mathbf{C} = \mathbf{c})$, $\bar{P}(Y = y \mid \mathbf{C}' = \mathbf{c}')$, then the empirical risks with respect to $\bar{\mathrm{SR}}(\mathbf{c}, \mathbf{c}')$ are defined as Eq. (13) below.

$$\bar{\mathrm{SR}}(\mathbf{c}, \mathbf{c}') := \mathbb{E}_{\mathcal{D}} \mathbb{E}_{\mathbf{c} \sim \bar{P}(\mathbf{C}|\mathbf{X}=\mathbf{x})} \bar{P}(Y = y \mid \mathbf{C} = \mathbf{c})$$
$$- \mathbb{E}_{\mathcal{D}} \mathbb{E}_{\mathbf{c} \sim \bar{P}(\mathbf{C}'|\mathbf{X}=\mathbf{x})} \bar{P}(Y = y \mid \mathbf{C}' = \mathbf{c}'). \quad (13)$$

To address the issue that the estimation of expected risk is not available, we propose Corollary 1 using the Variational Inference, Hoeffding inequality, Markov inequality, and Jensen inequality following (Yang et al., 2024) to ensure the expected risk can be upper bounded by the empirical risk.

**Corollary 1.** *Suppose prior distributions for the representations $\gamma_{\mathbf{C}} := P(\mathbf{C})$, $\gamma_{\mathbf{C}'} := P(\mathbf{C}')$. With probability $1 - \delta$ over the choice of samples $X \sim \mathcal{D}$, for every parameter*

combination, $\left|\mathrm{SR}(\mathbf{c}, \mathbf{c}') - \bar{SR}(\mathbf{c}, \mathbf{c}')\right|$ *is upper bounded by*

$$\mathcal{L}_{KL} = -log(\sigma_{\mathbf{C}}) + log(\sigma_{\mathbf{C}'}). \tag{14}$$

Corollary 1 indicates that the error between the empirical risk and expected risk can be minimized as the KL divergences decrease. Proof for Corollary 1 is provided in Appendix B.4.

### 3.4. Satisfaction of Causal Assumptions

In this section, we concern the satisfaction of causal assumptions, *Exogeneity* (Assumption 1) and *Monotonicity* (Assumption 2).

As for *Monotonicity*, it emphasizes the value of $y$ given that $\mathbf{c}$ should be larger than given any other intervened feature $\mathbf{c}'$. This objective can be formulated as:

$$\max P_t(Y = y | \mathbf{C} = \mathbf{c}) - P_t(Y = y | \mathbf{C} = \mathbf{c}'), \tag{15}$$

which is equal to minimizing the susceptibility risk. Therefore, we can naturally introduce the *Monotonicity* measurement into the estimation of susceptibility risk.

As for *Exogeneity*, we first rewrite the assumption using information theory (Kraskov et al., 2004).

**Proposition 2.** *(Mutual Information Equivalence) The Exogeneity condition in Assumption 1 can be equivalently represented with the Mutual Information $I(\cdot; \cdot)$:*

$$Pr(Y = k | \mathbf{C}, \mathbf{E} = e_{t/s}) = Pr(Y = k | \mathbf{C})$$
$$\Leftrightarrow \max_{\mathbf{x}_{ti} \in k} \mathbb{E} \, I(\mathbf{C}_{ti}; \mathcal{C}_s^k), \tag{16}$$

where $\mathbf{C}_{ti}$ represents sample features in the target domain $G(\mathbf{x}_{ti})$ and $\mathcal{C}_s^k$ represents the center point representation in the source domain $G(\mathbf{x}_{si})$. Both $\mathbf{C}_t$ and $\mathcal{C}^k$ belong to the $k$-th category. If $\Pr(Y = k | \mathbf{C}, \mathbf{E} = e_{t/s}) = \Pr(Y = k | \mathbf{C})$, then certain domain traits are unnecessary for identifying $Y$. Maximizing $\mathbb{E}_{\mathbf{x}_{ti} \in k} I(\mathbf{C}_{ti}; \mathcal{C}_s^k)$ mitigates domain changes on causal features, ensuring resilient and trustworthy representation. Proposition 2 is proven in Appendix B.5.

To achieve $\mathbb{E}_{\mathbf{x}_{ti} \in k} I(\mathbf{C}_{ti}; \mathcal{C}_s^k)$, we introduce an innovative and efficient pseudo-labeling strategy termed *Virtual Multi-unknown-categories Prototype pseudo-labeling (VMP)* for classifying samples from the target domain, subsequently maximizing the feature information shared between the target domain and source domain through *Contrastive-inspired Feature Alignment (CFA)*.

**Virtual Multi-unknown-categories Prototype pseudo-labeling (VMP).** Unlike previous clustering-based algorithms, which group unknown classes into a unique class, this paper presupposes the existence of $U$ unknown classes in the representation space. The fundamental concept of this approach is to initially identify positive instances of recognized classes using confidence scores and compute positive

prototypes by these positive instances. Subsequently, these affirmative instances of known classes are eliminated, and the remaining samples are subjected to a clustering algorithm in order to estimate the prototypes of $U$ unknown classes. The pseudo-labels for the samples are determined by calculating and comparing the distances between each target domain sample and all prototypes.

Specifically, given the scores of $\mathbf{x}_t$ after the softmax function $\xi(h(G(\mathbf{x}_t)))$, for the $k$-th known class, we first define the Top-**M** $\xi k(h(G(\mathbf{x}_t)))$ scores represented as positive instances $\mathcal{P}^k$, and let the positive prototype representation $\mathrm{p}_t^k$:

$$\{\mathrm{p}_t^k\}^K = \{\frac{1}{\mathbf{M}} \sum_{\mathcal{P}_k} G(\mathbf{x}_{ti})\}_{k=1}^K. \tag{17}$$

For the unknown classes, we remove all positive instances of known classes and use the K-means algorithm to generate $U$ negative prototypes for the unknown classes.

$$\{\mathrm{p}_t^u\}^U = \underset{\mathbf{x}_t \in \mathcal{D}_t \setminus \{\mathcal{P}^k\}_{k=1}^K}{\text{K-means}} (G(\mathbf{x}_t)). \tag{18}$$

Here we set $U = K$, $M = N_t/(U + K)$. For sample $\mathbf{x}_{ti}$, the $o$-th element of its pseudo label $\hat{y}$ is defined as follows:

$$\hat{y}_o = \begin{cases} 1, \text{ if } s\left(G\left(\mathbf{x}_{ti}\right), \mathrm{p}_t^o\right) = \max\left\{s\left(G\left(\mathbf{x}_{ti}\right), \mathrm{p}_t^a\right)\right\}_{a=1}^{K+U} \\ \\ 0, \text{ if } s\left(G\left(\mathbf{x}_{ti}\right), \mathrm{p}_t^o\right) < \max\left\{s\left(G\left(\mathbf{x}_{ti}\right), \mathrm{p}_t^a\right)\right\}_{a=1}^{K+U} \end{cases}, \tag{19}$$

where $s(a, b)$ measures the similarity between $a$ and $b$. Based on Eq. (19), we obtain the pseudo labels $\hat{y}_k$ for all categories $o \in \{\mathcal{Y}_s, \mathcal{U}\} = \{1, ...., K, K + 1, ..., K + U\}$. Then, the samples with pseudo label $\{K + 1, ..., K + U\}$ are defined as unknown class samples.

**Contrastive-inspired Feature Alignment (CFA).** To fully leverage the labeled knowledge from the source domain, this paper adopts a contrastive learning strategy to align the feature space of shared classes across domains. Unlike the contrastive learning strategy in CSDA, the prototypes for comparison in this work include not only the source domain prototypes $\mathbf{p}_s^k$ but also the virtual unknown class prototypes $\mathbf{p}_t^u$ (estimated by Eq. (18)). The prototypes of known classes in the source domain are defined as:

$$\{\mathrm{p}_s^k\}^K = \{\frac{1}{n_s^k} \sum_{i=1}^{n_s^k} G(\mathbf{x}_{si})\}_{k=1}^K. \tag{20}$$

To prevent negative effects from pseudo label noise, the target domain samples used for contrastive learning are filtered by $I(\mathbf{x}_t)$: (i) the pseudo label obtained based on the highest logit is aligned with the label obtained through the VMP strategy, and (ii) the value of the highest logit is higher than 0.7.

$$I(\mathbf{x}_t) = \begin{cases} 1, \max_k \{\xi_k(h(G(\mathbf{x}_t)))\}_{k=1}^K \wedge \hat{y}_k = 1 \\ 0, \text{else} \end{cases}. \tag{21}$$

*Table 1.* Comparison results (%) of *Image-CLEF*. (Best in **bold** and second best in underline)

| Method | Average | | | B→C | | | B→I | | | B→P | | | C→B | | | C→I | | |
|---|---|---|---|---|---|---|---|---|---|---|---|---|---|---|---|---|---|---|
| | OS* | UNK | **HOS** | OS* | UNK | **HOS** | OS* | UNK | **HOS** | OS* | UNK | **HOS** | OS* | UNK | **HOS** | OS* | UNK | **HOS** |
| OSNN (Mendes et al., 2017) | 77.0 | 50.4 | 60.4 | 92.3 | 63.0 | 74.9 | 79.6 | 61.0 | 69.1 | 68.3 | 59.3 | 63.5 | 65.3 | 47.3 | 54.9 | 84.3 | 47.0 | 60.4 |
| OSBP (Saito et al., 2018) | 74.4 | 70.2 | 71.5 | 87.0 | 81.0 | 83.9 | 85.3 | 65.7 | 74.3 | 66.3 | 66.7 | 66.5 | 62.0 | 58.0 | 59.9 | 89.0 | 80.0 | 84.3 |
| STA (Liu et al., 2019) | 81.3 | 55.1 | 65.0 | 93.3 | 51.7 | 66.5 | 86.0 | 60.7 | 71.2 | 77.7 | 48.7 | 59.8 | 61.3 | 69.7 | 65.2 | 91.7 | 66.7 | 77.2 |
| ROS (Bucci et al., 2020) | 69.9 | 76.9 | 73.1 | 78.3 | 90.0 | 83.8 | 73.0 | 76.3 | 74.6 | 59.0 | 68.3 | 63.3 | 59.0 | 68.3 | 63.3 | 78.3 | 83.0 | 80.6 |
| DAOD (Fang et al., 2020) | 71.1 | 75.8 | 73.3 | 79.4 | 82.0 | 80.7 | 78.4 | 90.9 | 84.3 | 72.1 | 80.8 | 76.3 | 51.3 | 47.1 | 49.1 | 79.0 | 88.6 | 83.6 |
| ANNA (Li et al., 2023) | 78.2 | 85.6 | 81.4 | 95.3 | 98.3 | 96.8 | 81.3 | 84.7 | 83.0 | 74.0 | 75.0 | 74.5 | 58.0 | 83.0 | 68.3 | 87.0 | 93.0 | 89.9 |
| **Ours** | 81.0 | 88.3 | 84.3 | 95.0 | 97.9 | 96.4 | 87.4 | 88.9 | 88.1 | 78.5 | 82.5 | 80.5 | 61.5 | 80.5 | 69.7 | 90.8 | 93.8 | 92.3 |

| Method | C→P | | | I→B | | | I→C | | | I→P | | | P→B | | | P→C | | | P→I | | |
|---|---|---|---|---|---|---|---|---|---|---|---|---|---|---|---|---|---|---|---|---|---|
| | OS* | UNK | **HOS** | OS* | UNK | **HOS** | OS* | UNK | **HOS** | OS* | UNK | **HOS** | OS* | UNK | **HOS** | OS* | UNK | **HOS** | OS* | UNK | **HOS** |
| OSNN | 75.3 | 46.3 | 57.3 | 62.0 | 41.6 | 49.8 | 92.0 | 41.3 | 57.0 | 81.3 | 40.6 | 54.2 | 55.0 | 49.6 | 52.2 | 86.0 | 55.3 | 67.3 | 82.0 | 52.6 | 64.1 |
| OSBP | 87.7 | 53.7 | 66.7 | 55.7 | 60.7 | 58.1 | 80.7 | 92.7 | 86.3 | 66.3 | 74,3 | 70.1 | 52.3 | 61.0 | 56.3 | 94.0 | 68.0 | 78.9 | 66.0 | 80.7 | 72.6 |
| STA | 84.0 | 54.0 | 65.7 | 62.3 | 54.0 | 57.9 | 94.0 | 53.7 | 68.4 | 80.7 | 59.0 | 68.2 | 61.3 | 43.7 | 51.0 | 93.7 | 47.7 | 63.2 | 90.0 | 51.0 | 65.1 |
| ROS | 68.7 | 78.7 | 73.3 | 58.0 | 59.7 | 58.8 | 88.7 | 92.7 | 90.6 | 78.0 | 76.0 | 77.0 | 47.3 | 59.3 | 52.7 | 71.3 | 90.3 | 79.7 | 79.7 | 81.3 | 80.5 |
| DADO | 74.5 | 78.9 | 76.7 | 54.5 | 56.9 | 55.7 | 80.3 | 82.0 | 81.2 | 73.3 | 80.8 | 76.9 | 51.7 | 51.0 | 51.3 | 79.0 | 82.0 | 80.5 | 79.6 | 86.6 | 83.9 |
| ANNA | 78.7 | 84.0 | 81.2 | 56.0 | 78.0 | 65.2 | 94.3 | 97.7 | 96.0 | 80.7 | 82.7 | 81.7 | 54.0 | 73.7 | 62.3 | 94.0 | 93.7 | 93.8 | 85.0 | 83.3 | 84.2 |
| **Ours** | 80.9 | 84.5 | 82.7 | 60.8 | 79.4 | 68.9 | 94.5 | 97.9 | 96.2 | 82.4 | 87.2 | 84.7 | 58.9 | 78.2 | 67.2 | 94.0 | 97.0 | 95.5 | 87.2 | 91.7 | 89.4 |

The optimization objective of our proposed contrastive learning is

$$\mathcal{L}_{exo} = -\sum_{k=1}^{K}\sum_{i}^{n_t^k} \frac{s(G(\mathbf{x}_{ti}), \mathbf{p}_s^k) \times I(\mathbf{x}_{ti})}{S(G(\mathbf{x}_{ti}), \{\mathbf{p}_s^k\}^K) + S(G(\mathbf{x}_{ti}), \{\mathbf{p}_t^u\}^U)}, \quad (22)$$

where

$$S(G(\mathbf{x}_{ti}), \{\mathbf{p}_s^k\}^K) = \sum_{k=1}^{K} s(G(\mathbf{x}_{ti}), \mathbf{p}_s^k), \quad (23)$$

$$S(G(\mathbf{x}_{ti}), \{\mathbf{p}_t^u\}^U) = \sum_{u=K+1}^{K+U} s(G(\mathbf{x}_{ti}), \mathbf{p}_t^u). \quad (24)$$

where $s(a, b)$ denotes the similarity between $a$ and $b$. $n_t^k$ denotes the number of samples in the target domain with label $k$.

Our proposed contrastive-inspired $\mathcal{L}_{exo}$ comprises two effects: (i) For the source-share (known) categories, $\mathcal{L}_{exo}$ operates the domain alignment with intra-class compactness and inter-class separability; (ii) for the target-private (unknown) categories, $\mathcal{L}_{exo}$ increases the discrepancy between private-class samples and the source domain features, enabling the model to better recognize unknown classes.

### 3.5. Model Optimization

During the adaptation stage of the proposed **COSDA**, we implement the optimization objective as follows:

$$\mathcal{L} = \mathcal{L}_t + \lambda_s \mathcal{L}_s + \lambda_{exo} \mathcal{L}_{exo} + \mathcal{L}_{KL} + \mathcal{L}_{STA}, \quad (25)$$
Eq. (11)   Eq. (12)      Eq. (22)   Eq. (14)   Eq. (3)

where $\mathcal{L}_t + \lambda_s \mathcal{L}_s + \mathcal{L}_{KL}$ is from the OSDA theory framework. Intuitively, $\mathcal{L}_s$ measures the empirical susceptibility risk of the source domain using source domain labels, while $\mathcal{L}_t$ measures the empirical susceptibility risk of the target

domain using predicted pseudo-labels. $\mathcal{L}_{KL}$ is used to reduce the error between the expected risk and the empirical risk. $\mathcal{L}_{exo}$ is for satisfaction of the *Exogeneity* assumption, and $\mathcal{L}_{STA}$ tends to sufficiently large intervention operations to ensure that the learning parameters of the model are stable.

## 4. Experiments.

In this section, we verify the effectiveness of COSDA using real-world and synthetic datasets.

### 4.1. Experiment Setup

**Benchmark Dataset Settings.** Extensive experiments are conducted on five benchmarks following the standard setting (Qu et al., 2023; Bucci et al., 2020): *Office-Home* (Venkateswara et al., 2017), *Office-31* (Wang et al., 2019), *Image-CLEF* (Li et al., 2023), *DomainNet* (Wen & Brbic, 2024), and *VisDA* (Li et al., 2021).

**Evaluation Metrics.** Following the main OSDA stream (Qu et al., 2023; Li et al., 2023), we utilize four widely used measures, i.e., **UNK**, **OS***, **OS**, and harmonic mean accuracy (**HOS**). Among them, **HOS** is regarded as the most equitable evaluation criterion.

$$HOS = \frac{2 \times OS^* \times UNK}{OS^* + UNK}. \quad (26)$$

**Implementation Details.** We adopt the same network architecture as mainstream OSDA methods (Qu et al., 2023; Li et al., 2023). All experiments are conducted using the ImageNet (Deng et al., 2009) pre-trained ResNet-50 (He et al., 2016) feature extractor. First, we train a source model on the source domain. During target model adaptation, we apply the SGD optimizer with a momentum of 0.9 and a batch size of 32 for all benchmark datasets, following (Li et al., 2023). We set the learning rate to $1 \times 10^{-3}$ for *Office-31*, *Image-CLEF*, and *Office-Home*. For hyperparameters, $\lambda_s$=0.2, $\lambda_{exo}$=1. All experiments are conducted

*Table 2.* Comparison results (%) of *Office-31*. (Best in **bold** and second best in underline)

| Method | Average | | | A→D | | | A→W | | | D→A | | | D→W | | | W→A | | | W→D | | |
|---|---|---|---|---|---|---|---|---|---|---|---|---|---|---|---|---|---|---|---|---|---|
| | OS* | UNK | **HOS** | OS* | UNK | **HOS** | OS* | UNK | **HOS** | OS* | UNK | **HOS** | OS* | UNK | **HOS** | OS* | UNK | **HOS** | OS* | UNK | **HOS** |
| OSBP (Saito et al., 2018) | 87.2 | 80.4 | 83.7 | 90.5 | 75.5 | 82.4 | 86.8 | 79.2 | 82.7 | 76.1 | 72.3 | 75.1 | 97.7 | 96.7 | 97.2 | 73.0 | 74.4 | 73.7 | 99.1 | 84.2 | 91.1 |
| STAsum (Liu et al., 2019) | **94.6** | 50.5 | 65.5 | 95.4 | 45.5 | 61.6 | 92.1 | 58.0 | 71.0 | **94.1** | 55.0 | 69.4 | 97.1 | 49.7 | 65.5 | 92.1 | 46.2 | 60.9 | 96.6 | 48.5 | 64.4 |
| UAN (You et al., 2019) | 93.4 | 40.3 | 55.1 | **95.6** | 24.4 | 38.9 | **95.5** | 31.0 | 46.8 | 93.5 | 53.4 | 68.0 | **99.8** | 52.5 | 68.8 | **94.1** | 38.8 | 54.9 | 81.5 | 41.4 | 53.0 |
| ROS (Bucci et al., 2020) | 86.6 | 85.8 | 85.9 | 87.5 | 77.8 | 82.4 | 88.4 | 76.7 | 82.1 | 74.8 | 81.2 | 77.9 | 99.3 | 93.0 | 96.0 | 69.7 | 86.6 | 77.2 | **100.0** | 99.4 | 99.7 |
| ANNA (Li et al., 2023) | 87.8 | 90.0 | 88.6 | 93.2 | 76.1 | 83.8 | 82.8 | 88.4 | 85.5 | 75.4 | 91.1 | 82.5 | 99.4 | 99.6 | 99.5 | 76.0 | 87.9 | *81.6* | **100.0** | 96.8 | 98.4 |
| GLC (Qu et al., 2023) | - | - | 89.0 | - | - | - | - | - | - | - | - | - | - | - | - | - | - | - | - | - | - |
| OSLPP (Wang et al., 2024) | 89.3 | 85.6 | 87.4 | 92.6 | 90.4 | 91.5 | 89.5 | 88.4 | 89.0 | 82.1 | 76.6 | 79.3 | 96.9 | 88.0 | 92.3 | 78.9 | 78.5 | 78.7 | 95.8 | 91.5 | 93.6 |
| GLC++ (Qu et al., 2024) | - | - | *90.4* | - | - | - | - | - | - | - | - | - | - | - | - | - | - | - | - | - | - |
| **Ours** | 91.4 | **93.9** | **92.6** | 91.3 | 85.9 | **88.6** | 86.7 | **91.5** | **89.1** | 90.2 | **96.9** | **93.4** | 92.1 | **98.8** | 95.3 | 91.5 | **94.7** | **93.1** | 96.6 | 95.3 | 95.9 |

*Table 3.* Ablation study results (%) on *Office-Home* with four different sub-tasks. (Best in **bold** and second best in underline)

| Office-Home | A-R | | | | R-P | | | | P-C | | | | C-A | | | | Average | | | |
|---|---|---|---|---|---|---|---|---|---|---|---|---|---|---|---|---|---|---|---|---|
| | OS* | UNK | OS | **HOS** | OS* | UNK | OS | **HOS** | OS* | UNK | OS | **HOS** | OS* | UNK | OS | **HOS** | OS* | UNK | OS | **HOS** |
| *Causal Assumption* | | | | | | | | | | | | | | | | | | | | |
| w/o $\mathcal{L}_{exo}$ | 75.6 | 80.0 | 75.8 | 77.7 | 73.6 | 82.6 | 74.0 | 77.8 | 56.9 | 74.8 | 57.6 | **64.6** | 52.2 | 75.0 | 53.1 | 61.6 | 64.6 | 78.1 | 65.1 | 70.4 |
| *OSDA Theory Framework* | | | | | | | | | | | | | | | | | | | | |
| w/o $\mathcal{L}_t$ | **89.0** | 0.0 | **85.6** | 0.0 | 82.9 | 0.0 | 79.7 | 0.0 | 49.8 | 0.0 | 47.9 | 0.0 | 61.5 | 0.0 | 59.1 | 0.0 | 70.8 | 0.0 | 68.1 | 0.0 |
| w/o $\mathcal{L}_s$ | 72.9 | **85.2** | 73.4 | 78.6 | 71.1 | **87.2** | 71.7 | 78.3 | 51.5 | **77.9** | 52.5 | 62.0 | 49.8 | **77.6** | 50.9 | 60.7 | 64.6 | **82.0** | 62.1 | 69.9 |
| *Pseudo labelling Strategy* | | | | | | | | | | | | | | | | | | | | |
| with MLP | 84.3 | 0.0 | 81.1 | 0.0 | **83.1** | 0.0 | 79.9 | 0.0 | 55.7 | 0.0 | 53.6 | 0.0 | **66.6** | 0.0 | **64.1** | 0.0 | **72.4** | 0.0 | **69.7** | 0.0 |
| with GC | 77.2 | 82.9 | 77.2 | 77.0 | 73.6 | 84.0 | 74.0 | 78.4 | 51.5 | 75.5 | 52.5 | 61.3 | 50.7 | 74.1 | 51.6 | 60.2 | 63.3 | 78.7 | 63.8 | 69.2 |
| **Ours** | 80.3 | 79.5 | 80.2 | **79.9** | 81.1 | 77.3 | 80.9 | **79.1** | 58.5 | 69.9 | 58.9 | 63.7 | 57.9 | 74.0 | 58.6 | **65.0** | 69.4 | 75.2 | 69.6 | **71.9** |

on an RTX-4090 GPU with PyTorch-1.10. We compare our method with recent works that adopt the same OSDA settings. All baseline scores are directly taken from their publicly reported results. A detailed review of the baselines is provided in Appendix C. The code is available at *https://github.com/ZHOURui6025/COSDA-master*.

## 4.2. Benchmark Comparison

***Image-CLEF***. We report the comparison results of the real-world benchmark *Image-CLEF* in Table 1. We observe that **COSDA** achieves the best results in 11 of 12 sub-tasks for the HOS comparison and 10 of 12 tasks for the UNK comparison, verifying our great potential for more complex real-world scenes. Moreover, **COSDA** achieves the best 84.3% average HOS, which yields 2.9% gains over the state-of-the-art work **ANNA**, 11.0% over **DADO** (Fang et al., 2020), 12.8% over **OSBP**, and the best 88.3% average UNK, which outperforms **ANNA** by 2.7%, **DADO** by 12.5%, and **OSBP** by 18.1%, verifying the effectiveness of our method.

***Office-31***. Comparison results on *Office-31* are shown in Table 2. The proposed method achieves the best average HOS (92.6%) over all 6 tasks, outperforming **OSBP** (Saito et al., 2018), **ROS** (Bucci et al., 2020), and **OSLPP** (Wang et al., 2024) with 8.9%, 6.7%, and 5.2% HOS and 13.5%, 8.1%, and 8.3% UNK, respectively. Compared with the state-of-the-art OSDA (source-free) work **GLC++** (Qu et al., 2024), our method comprehensively surpasses it with 2.2% HOS. Compared with the state-of-the-art OSDA (source-support) work **ANNA** (Li et al., 2023), our method comprehensively

surpasses it with 3.6% OS*, 3.9% UNK, and 4% HOS, respectively, verifying the effect of our method.

***Office-Home***. Comparison results on *Office-Home* are shown in Table 6. **COSDA** gives the best average HOS (71.6%) evaluated over 12 tasks. Specifically, our method outperforms **OSBP**, **ROS**, and **OSLPP** with 7.0%, 5.5%, and 4.7% HOS, and surpasses them by 8.0%, 1.9% and, 2.6% in UNK comparison, demonstrating the robustness of our unbiased OSDA framework.

## 4.3. Efficiency of Susceptibility Risk

In this section, we conduct synthetic experiments to evaluate the effectiveness of susceptibility risk. The effectiveness of susceptibility risk is demonstrated by examining whether it can learn the essential causal relationships. To this end, we designed a synthetic data generator based on Fig. 3 and created a sample set $\{\mathbf{X}, Y\}$, where $\mathbf{X}$ is generated from three causal factors $\mathbf{C}$ and a spurious association factor $\mathbf{S}$.

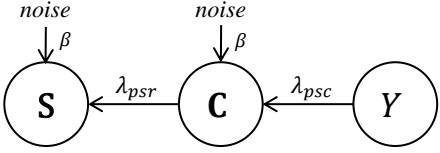

*Figure 3.* Causal graph for Synthetic Data Generation: $Y \perp S \mid C$

We use a parameter $\lambda_{\text{psc}}$ to distinguish the importance of causal factors. When $\lambda_{\text{psc}}$ is closer to 0 or 1, the causal association is stronger to Y; Conversely, the closer $\lambda_{\text{psc}}$ is to 0.5, the weaker the causal association is. Then we introduce

a parameter $\beta$ to represent the noise intensity in the data. We set $\lambda_{\text{psc}} = \{0.15, 0.25, 0.35\}$ and $\beta = \{0.1, 0.4, 0.7, 1.1\}$.

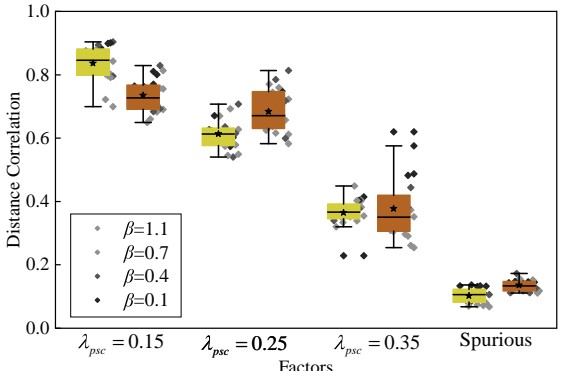

*Figure 4.* Results of Suscepetibility risk ( yellow ) and Vanilla risk ( orange )

Besides, $\lambda_{psr}$ denotes the spurious degree. When $\lambda_{psr}$ gets higher, the spurious correlation is stronger in data $X$. We set $\lambda_{psr} = 0.4$, and the dimension of the factors $d = 64$. Then we develop a non-linear function to generate $\mathbf{X}$ from $[C_{0.15}, C_{0.25}, C_{0.35}, S]$. We use Distance Correlation to evaluate the correlation between the learned representation and these factors. A higher Distance Correlation indicates a better representation of factors. In Fig. 4, we compare the performance of the susceptibility risk evaluator with that of a standard empirical risk evaluator. For critical causal factors ($\lambda_{psc} = 0.15$), our method achieved a distance correlation higher than the standard risk. For factors with lower causal relevance ($\lambda_{psc} = 0.25$, $\lambda_{psc} = 0.35$). For spurious factors, our method yielded a lower Distance Correlation compared to the standard risk. These results verified that by incorporating susceptibility risk evaluation, models can reduce the learning of spurious correlations and maintain a focus on critical causal factors. Due to page limitations, additional details and results regarding synthetic experiments are provided in the Appendix D.

## 4.4. Ablation Study

In this part, we conducted detailed ablation studies on four subtasks in *Office-Home* and summarized the efficiency of different parts of COSDA.

**Efficiency of *CFA*.** As shown in Table 3, compared to the complete COSDA, removing $\mathcal{L}_{exo}$ (row 3) results in a 1.5% decrease in average HOS, with a particularly notable 4.8% reduction in OS*. This indicates that the *CFA* significantly enhances the model's ability to recognize known categories.

**Efficiency of the Evaluation on Source/Target Risks.** As shown in Table 3, when $\mathcal{L}_t$ is removed (row 5), the model fails to recognize unknown classes. Conversely, when the $\mathcal{L}_s$ is removed (row 6), the model becomes overconfident

for unknown classes, while its ability to recognize known categories declines. Overall, removing the risk from either domain negatively impacts the model's performance, with the OS dropping by 1.5% and 7.5%, respectively.

**Efficiency of *VMP*.** To assess the effectiveness of *VMP*, we compare it to the logit-based pseudo-labeling strategy (MLP) (Wang et al., 2022) and the Global Clustering pseudo-labeling strategy (GC) (Qu et al., 2023; Saito, 2021). The comparison illustrates that MLP (row 8) has virtually no discriminative ability for unknown categories. This is most likely because it relies heavily on source domain knowledge and lacks unknown class information. Furthermore, our technique outperforms GC by 2.7%, demonstrating the usefulness of VMP.

## 4.5. Qualitative Results

**Trade-off Analysis.** To analyze the impact of parameters in Eq. (25) on COSDA performance, we define $\lambda_s = \{0.1, 0.2, 0.5, 0.8, 1.0\}$ and $\lambda_{exo} = \{0.1, 0.2, 0.5, 0.8, 1.0\}$. Fig. 5 shows that the proposed technique has good parameter stability across datasets.

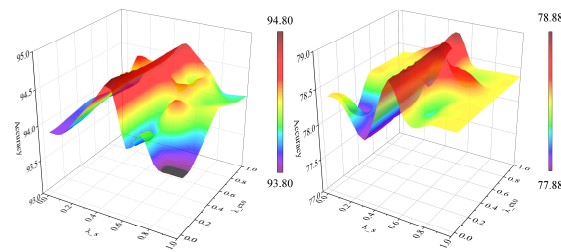

*Figure 5.* The sensitivity analysis ($\lambda_s$, $\lambda_{exo}$) of COSDA on P-I (*Image-CLEF*) task (left) and Rw-Pr (*Office-Home*) task (right) in terms of parameter variations.

**Feature-level Analysis.** To illustrate the success of our strategy, we compare the characteristics recovered by the baseline algorithms and our approach to the t-SNE. Fig. 6 shows the t-SNE visualization of feature distributions on the $W \to D$ task (*Office-31*, left) and $Ar \to Rw$ task (*Office-Home*, right) with the ResNet-50 backbone. Comparative methods include OSBP, ANNA, and COSDA. The gray node denotes the unknown target sample, the red node denotes the known source sample, and the blue node denotes the known target sample. Compared with OSBP, our method clusters class characteristics more compactly, which indicates the improvement of the decision border between known and unknown classes. Since both OSBP and ANNA utilize adversarial training and share the same underlying framework, their feature spaces exhibit similar characteristics. Comparatively, ANNA better delineates the boundaries between known and unknown classes, as only a small number of gray points overlap with the blue and red points. Due to their similar performance, we cannot clearly exhibit COSDA's advantage at the feature level compared with ANNA. However,

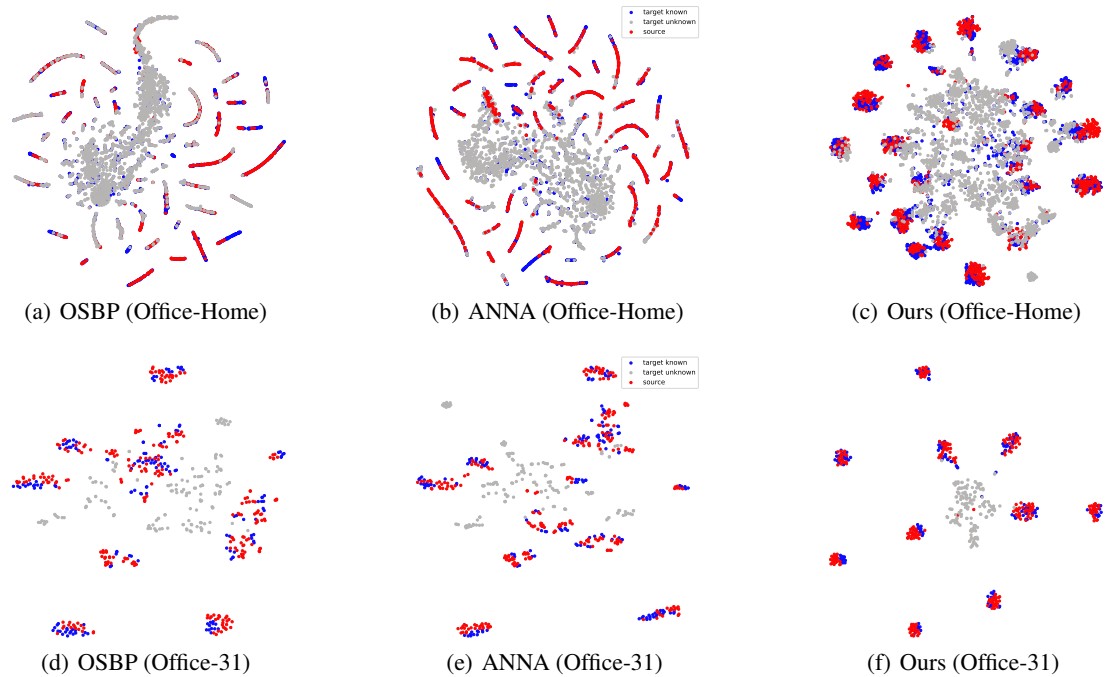

|  |  |  |
|---|---|---|
| (a) OSBP (Office-Home) | (b) ANNA (Office-Home) | (c) Ours (Office-Home) |
| (d) OSBP (Office-31) | (e) ANNA (Office-31) | (f) Ours (Office-31) |

*Figure 6.* The t-SNE visualization of feature distributions on the $W \rightarrow D$ task (*Office-31*, left) and $Ar \rightarrow Rw$ task (*Office-Home*, right) with the ResNet-50 backbone.

the clustering tendency of local unknown classes suggests that VMP's approach of considering unknown classes as multiple distinct groups is beneficial.

*Table 4.* Comparison results on DomainNet and VisDA with CLIP backbone (All baseline results are obtained from Reference (Deng & Jia, 2023)).

| Method | DomainNet(173/172) | | | VisDA(6/6) | | |
|---|---|---|---|---|---|---|
|  | OS* | UNK | HOS | OS* | UNK | HOS |
| DCC (Li et al., 2021) | 50.2 | 45.1 | 47.5 | 75.3 | 46.2 | 57.3 |
| UNIOT (Deng & Jia, 2023) | 59.2 | 45.1 | 51.2 | 75.7 | 49.4 | 59.8 |
| CROW (Wen & Brbic, 2024) | 70.3 | 50.9 | 59.0 | 77.0 | 62.8 | 69.2 |
| COSDA-CLIP | **72.0** | **76.2** | **73.9** | **85.2** | **72.6** | **78.4** |

**Comparison on more datasets and more backbones.** We implemented additional experiments on VisDA (Li et al., 2021) and DomainNet (Deng & Jia, 2023; Wen & Brbic, 2024) and discovered that our method demonstrates good performance on both CNN-based and CLIP-based architectures. Considering both time constraints and GPU memory demands (particularly for the larger models), we utilized six 40GB NVIDIA A100 GPUs to execute the new experiments. DomainNet and VisDA use the same hyperparameter settings as smaller-scale datasets, specifically $\lambda_s = 0.2$, $\lambda_{exo} = 1$. But the learning rate has been reduced, specifically $lr = 5e - 4$. On DomainNet, COSDA-CLIP achieves 73.9% HOS, outperforming CROW by 14.9%. On VisDA, COSDA-CLIP reaches 78.4% HOS, outperforming CROW by 9.2%. With CLIP, COSDA particularly further enhances known-class classification and unknown-class detection (UNK).

## 5. Conclusion

This paper establishes a theoretical framework for OSDA grounded in the classical concept of *Susceptibility* to address the OSDA problem. We further present two modules, designated as *CFA* and *VMP*, to fulfill causal assumptions and mitigate both feature alignment risk and open-set risk. Experiments show that COSDA outperforms the competition on all three benchmarks.

## Acknowledgements

This work was supported by the National Key Research and Development Program of China (Grant No. 2022YFB3305803). BH was supported by the NSFC General Program No. 62376235, GDST Basic Research Fund Nos. 2022A1515011652 and 2024A1515012399. Rui Zhou was supported by the National Natural Science Foundation of China under Grant 62276272, the Training Program for Excellent Young Innovators of Changsha under Grant KQ2009009, and the Postgraduate Scientific Research Innovation Project of Hunan Province under Grant XJJC2024022.

## Impact Statement

This paper presents work whose goal is to advance the field of Machine Learning. There are many potential societal consequences of our work, none of which we feel must be specifically highlighted here.

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

# —————Appendix—————

The structure of the Appendix is as follows

- Appendix A gives examples to illustrate where the susceptibility is high and where it is low.

- Appendix B contains all missing proofs in the main manuscript.

- Appendix C contains the extended related work.

- Appendix D details the dataset, implementation details and additional experimental results.

## A. Understanding Susceptibility

As a fundamental concept in causal inference, *susceptibility* pertains to determining whether an event (cause) leads to another event (effect). The average causal effect (ACE) is a classical metric for quantifying the strength of causal relationships, defined as:

$$ACE = P(B|A) - P(B|\neg A), \tag{27}$$

where the first term represents the probability of event $B$ occurring given that event $A$ occurs, and the second term measures the probability of $B$ occurring in the absence of $A$. By intervening on causal variable $A$ (i.e., altering their states), we can observe changes in the outcome $B$. In the complex physical world, one event often has multiple causes, and it is crucial to evaluate their causal significance. Generally, a larger $ACE$ indicates a stronger causal relationship, implying a more significant influence of the cause on the effect and greater mutual information between the cause and the effect.

The probability of susceptibility (PS) is defined as:

equal to $ACE$

$$PS = \frac{P(B|A) - P(B|\neg A)}{1 - P(B|\neg A)}. \tag{28}$$

the association between $A$ and $B$ after intervention

Here, the numerator is identical to Eq. 27, while the denominator quantifies the association between events $A$ and $B$ after intervention. When interventions of the same magnitude on $A$ produce equivalent changes in $B$, the average causal effect treats them as equal. However, a higher $P(B \mid \neg A)$ indicates stronger susceptibility, guiding us to pay greater attention to it, which is intuitively more reasonable. We provided three examples to illustrate the effectiveness of *susceptibility*.

For the label "maple", we define four relevant characteristics: (1) *palmately lobed leaves*, (2) larger leaves with a few *prominent teeth*, (3) *palmate veins*, and (4) *orange or red* color. The conditional probabilities between these characteristics and the label (i.e., whether a leaf belongs to a maple tree) are summarized in Fig. 7. When comparing a certain feature, we presume that information regarding other features is inaccessible.

**Example 1. (with same $ACE$)** The characteristic "palm-shaped lobes" is regarded as the most distinguishing feature of maple leaves, whose presence firmly signifies a maple leaf. Conversely, the distinguishing of "palmate veins" is another relative trait of maple leaves, albeit less pertinent than "palm-shaped lobes." Given these assumptions, the average causal effect (ACE) for both features is determined to be identical, at 0.5. Nonetheless, susceptibility risk directs the model to emphasize the characteristic "*palm-shaped lobes*" owing to its elevated $P(B|A)$. This underscores the superiority of the susceptibility model in detecting more distinctive causal factors.

**Example 2. (with same $P(B|\neg A)$ )** For the label "maple", the feature "*orange or red*" is not causally relative to "maple". Therefore, given $C = 0$ or $C = 1$, the conditional probability $P(Y|C)$ remains unchanged, so we assume it to be 0.3 in both cases. Under these assumptions, $P(B|\neg A)$ is equal for "*orange or red*" and "*palace-shaped lobes*". Relying solely on $P(B|\neg A)$ may lead to misguided attention, as it fails to distinguish between causally relevant and irrelevant features. By calculating the $PS \approx 0.43$, we find that the feature "*prominent teeth*" is more significant than "*orange or red*", which aligns with the results of comparing $ACE$.

| label ($Y$) | feature ($\mathbf{C}$) | $P(Y\vert\mathbf{C}=1)$ | $P(Y\vert\mathbf{C}=0)$ | $ACE$ | $PS$ |
|---|---|---|---|---|---|
| maple  |  palm-shaped lobes |  $P(Y\vert\mathbf{C}=1)=0.8$ |  $P(Y\vert\mathbf{C}=0)=0.3$ | $ACE=0.8-0.3$ $=0.5$ | $PS=\dfrac{0.8-0.3}{1-0.3}$ $\approx 0.7$ |
| |  prominent teeth |  $P(Y\vert\mathbf{C}=1)=0.6$ |  $P(Y\vert\mathbf{C}=0)=0.3$ | $ACE=0.6-0.3$ $=0.3$ | $PS=\dfrac{0.6-0.3}{1-0.3}$ $\approx 0.43$ |
| |  palmate veins |  $P(Y\vert\mathbf{C}=1)=0.6$ |  $P(Y\vert\mathbf{C}=1)=0.1$ | $ACE=0.6-0.1$ $=0.5$ | $PS=\dfrac{0.6-0.1}{1-0.1}$ $\approx 0.56$ |
| |  orange, or yellow |  $P(Y\vert\mathbf{C}=1)=0.3$ |  $P(Y\vert\mathbf{C}=0)=0.3$ | $ACE=0.3-0.3$ $=0$ | $PS=\dfrac{0.3-0.3}{1-0.3}$ $=0$ |

Figure 7. Examples for understanding susceptibility in image classification problem

**Example 3. (with same $P(B\vert A)$ )** The feature "*prominent teeth*" is assumed to be a third-relative characteristic of maple leaves. If a leaf lacks "*prominent teeth*", it is highly unlikely to be a maple leaf. However, the presence of "*prominent teeth*" does not exclusively indicate a maple leaf, as other species may also exhibit this trait. Therefore, we assume $P(Y=1\vert\mathbf{C}=1)=0.6$, $P(Y=1\vert\mathbf{C}=1)=0.3$. Under these assumptions, $P(B\vert A)$ is equal for "*prominent teeth*" and "*palace-shaped lobes*". However, susceptibility risk guides the model to prioritize the feature "*palm-shaped lobes*", which exhibits a greater difference than $ACE$. This result highlights the superiority of the susceptibility model in identifying more discriminative causal features.

In summary, susceptibility provides a more robust measure of the causal relationship between features and the target. This work proposes a susceptibility risk estimator derived from this principle, aiming to guide the model in learning more causally relevant features and improving its generalization capabilities across domains.

**The connections of SRE, CFA, and VMP.** SRE quantifies causal feature representation ability via susceptibility analysis. Direct SRE estimation in the target domain is infeasible due to label scarcity. To resolve this problem, we first decompose target-domain expected risk into open-set and closed-set, then bridge source/target risks using domain-invariant representations, and finally derive generalization bounds to narrow the gap between empirical and expected susceptibility risks (Theorems 1 2 & Corollary 1). To satisfy the exogeneity assumption for causal identifiability, we propose CFA, which encourages independence across causal features belonging to different categories via information bottleneck (Proposition 2) and then introduces the VMP pseudo-label strategy.

## B. Proofs

**Assumption 1** ((Exogeneity (Causality (Pearl, 2009)))). *The Exogeneity of C holds if the following invariant conditions are satisfied in Fig. 1(b): $\mathbf{E}\perp\mathbf{C}$, $\mathbf{E}\perp Y\mid\mathbf{C}$, which can be equivalently represented using the probability: $Pr(Y\vert\mathbf{C},\mathbf{E}=\mathrm{e}_{t/s})=Pr(Y\vert\mathbf{C})$, where $e_{t/s}$ is the domain-specific feature.*

**Assumption 2** ((Monotonicity (Causality (Pearl, 2009)))). *If the causal feature of y is c, Y is monotonic relative to $\mathbf{X}$ if and only if $y'_{\mathbf{c}}\wedge y_{\mathbf{c}'}=false$.*

In this section, we provided theoretical proofs. Theorem 2 and Corollary 1 follow the PAC-Bayesian theory framework (Catoni, 2007; Germain et al., 2016). Proposition 2 is based on Mutual Information theorey (Kraskov et al., 2004). The proof of Monotonicity is built upon Causality theory (Pearl, 2009).

**Algorithm 1** Overall Training Process of COSDA

---

**Input:** source domain $\mathcal{D}_S$, target domains $\mathcal{D}_T$; pretrained source model $f_\theta^0$, number of epochs $E$, batch size $B$, Trade-off hyperparameters $\lambda_s$, $\lambda_{exo}$, a blank Memory $\mathcal{M}$.

Initialize parameters randomly; $\mathcal{M} \leftarrow f^0$.

**for** $epoch = 1$ **to** $E$ **do**

    Compute prototypes $\mathbf{p}_t^{k/u}$ by VMP (Eq. (17), (18)).

    **for** $b = 1$ **to** $\frac{\max\{|\mathcal{D}_s|,|\mathcal{D}_t|\}}{B}$ **do**

        Sample a mini-batch $B_s = \{\mathbf{X}_{sb}, \mathbf{y}_{sb}\}$, $B_t = \{\mathbf{X}_{tb}\}$ from $\mathcal{D}_S, \mathcal{D}_T$.

        Forward pass: $\hat{\mathbf{y}}_s^{\mathbf{c}}, \hat{\mathbf{y}}_s^{\mathbf{c}'}, \sigma_s^{\mathbf{c}}, \sigma_s^{\mathbf{c}'}, \mathbf{Z}_s^c, \text{int}_s = f(B_s; \theta)$, where $int_s$ denotes the intervention on $\mathbf{Z}_s^c$.

        Compute loss: $\mathcal{L}_s, \mathcal{L}_{\text{KL}(s)}, \mathcal{L}_{\text{STA}(s)} \leftarrow \mathbf{y}_{sb}, \hat{\mathbf{y}}_s^{\mathbf{c}}, \hat{\mathbf{y}}_s^{\mathbf{c}'}, \sigma_s^{\mathbf{c}}, \sigma_s^{\mathbf{c}'}, \text{int}_s$. (Eq. (12), (14), (3))

        Get pesudo labels $\tilde{\mathbf{y}}_{tb}$ of $B_t$ by VMP with $\mathbf{p}_t^{k/u}$. (Eq. (19))

        Forward pass: $\hat{y}_t^{\mathbf{c}}, \hat{y}_t^{\mathbf{c}'}, \sigma_t^{\mathbf{c}}, \sigma_t^{\mathbf{c}'}, \mathbf{Z}_t^c, int_t = f(B_t; \theta)$.

        Compute loss: $\mathcal{L}_t, \mathcal{L}_{\text{KL}(t)}, \mathcal{L}_{\text{STA}(t)} \leftarrow \tilde{\mathbf{y}}_{tb}, \hat{\mathbf{y}}_t^{\mathbf{c}}, \hat{\mathbf{y}}_t^{\mathbf{c}'}, \sigma_t^{\mathbf{c}}, \sigma_t^{\mathbf{c}'}, \text{int}_t$. (Eq. (11), (14), (3))

        Update $\mathcal{M} \leftarrow \mathbf{Z}_s^c$.

        Get $\mathbf{p}_s^k$ by $\mathcal{M}$. (Eq. (21))

        Compute loss: $\mathcal{L}_{exo} \leftarrow \tilde{\mathbf{y}}_{tb}, \mathbf{p}_s^k, \mathbf{p}_t^u$. (Eq. (22))

        Calculate the overall loss, as $\mathcal{L} \leftarrow$ Eq. (25)

        Backward pass: compute gradients $\nabla_\theta \mathcal{L}$.

        Update parameters: $\theta = \theta - \eta \nabla_\theta \mathcal{L}$.

    **end for**

**end for**

**Output:** Final Optimized Models $f_\theta^*$.

---

## B.1. Proof of Proposition 1

*Proof.* Based on Eq. (5) and Eq. (28), $PS$ can be altered by Eq. (29) below.

$$PS_t = \frac{POR_t - PIR_t}{1 - PIR_t} \tag{29}$$

Then,

$$
\begin{aligned}
\ln(\text{SR}_t(\mathbf{c}, \mathbf{c}')) &= 1 - \frac{POR_t - PIR_t}{1 - PIR_t} \\
&= \ln(1 - PS) \\
&= \ln(\frac{1 - POR_t}{1 - PIR_t}) \\
&= \ln(1 - POR_t) - \ln(1 - PIR_t) \\
&= \ln(1 - \mathbb{E}_{(\mathbf{x},y)\sim\mathcal{T}}\mathbb{E}_{\mathbf{c}\sim P_t(\mathbf{C}|\mathbf{X}=\mathbf{x})}P(Y = y \mid \mathbf{C} = \mathbf{c})) \\
&\quad - \ln(1 - \mathbb{E}_{(\mathbf{x},y)\sim\mathcal{T}}\mathbb{E}_{\mathbf{c}'\sim P_t(\mathbf{C}'|\mathbf{X}=\mathbf{x})}P(Y = y \mid \mathbf{C}' = \mathbf{c}') \\
&= \ln(\mathbb{E}_{(\mathbf{x},y)\sim\mathcal{T}}\mathbb{E}_{\mathbf{c}\sim P_t(\mathbf{C}|\mathbf{X}=\mathbf{x})}P(Y \neq y \mid \mathbf{C} = \mathbf{c})) \\
&\quad - \ln(\mathbb{E}_{(\mathbf{x},y)\sim\mathcal{T}}\mathbb{E}_{\mathbf{c}'\sim P_t(\mathbf{C}'|\mathbf{X}=\mathbf{x})}P(Y \neq y \mid \mathbf{C}' = \mathbf{c}') \\
&= \ln(O_t(\mathbf{c})) - \ln(I_t(\mathbf{c}'))
\end{aligned}
\tag{30}
$$

Then,

$$
\begin{aligned}
\text{SR}_t(\mathbf{c}, \mathbf{c}') &= e^{\ln(O_t(\mathbf{c})) - \ln(I_t(\mathbf{c}'))} \\
&< e^{O_t(\mathbf{c}) - I_t(\mathbf{c}') - 2} \\
&< e^{O_t(\mathbf{c}) - I_t(\mathbf{c}')}
\end{aligned}
\tag{31}
$$

Then, by minimizing $O_t(\mathbf{c}) - I_t(\mathbf{c}')$, we can minimize $\text{SR}_t(\mathbf{c}, \mathbf{c}')$. Therefore, we can get the result of Proposition 1. By minimizing $\text{SR}_t(\mathbf{c}, \mathbf{c}')$, it can improve the consistency of the representation results of the model.

$\square$

## B.2. Proof of Theorem 1

Given a domain $\mathcal{S}/\mathcal{T}$, an invariant representation inference model $G$, an intervention model $G'$, a classifier $h$,

$$
\begin{aligned}
\mathrm{SR}_{\mathcal{T}}(\mathbf{c}, \mathbf{c}') &= O_t(\mathbf{c}) - I_t(\mathbf{c}') \\
&= \mathop{\mathbb{E}}_{(\mathbf{x},y)\sim\mathcal{T}} \mathop{\mathbb{E}}_{\mathbf{c}\sim P_t^G(\mathbf{C}|\mathbf{X}=\mathbf{x})} P^h(Y \neq y \mid \mathbf{C} = \mathbf{c}) \\
&\quad - \mathop{\mathbb{E}}_{(\mathbf{x},y)\sim\mathcal{T}} \mathop{\mathbb{E}}_{\mathbf{c}'\sim P_t^{G'}(\mathbf{C}'|\mathbf{X}=\mathbf{x})} P^h(Y \neq y \mid \mathbf{C}' = \mathbf{c}') \\
&= \mathop{\mathbb{E}}_{(\mathbf{x},y)\sim\mathcal{T}}[I(\mathbf{x}, y) \in \mathrm{supp}(\mathcal{S})] \left[ \mathop{\mathbb{E}}_{\mathbf{c}\sim P_t^G(\mathbf{C}|\mathbf{X}=\mathbf{x})} P^h(Y \neq y \mid \mathbf{C} = \mathbf{c}) \right. \\
&\qquad\qquad\qquad\qquad\qquad\qquad\qquad \left. - \mathop{\mathbb{E}}_{\mathbf{c}'\sim P_t^{G'}(\mathbf{C}'|\mathbf{X}=\mathbf{x})} P^h(Y \neq y \mid \mathbf{C}' = \mathbf{c}') \right] \\
&\quad + \mathbb{E}_{(\mathbf{x},y)\sim\mathcal{T}}[I(\mathbf{x}, y) \notin \mathrm{supp}(\mathcal{S})] \left[ \mathop{\mathbb{E}}_{\mathbf{c}\sim P_t^h(\mathbf{C}|\mathbf{X}=\mathbf{x})} P^D(Y \neq K + 1 \mid \mathbf{C} = \mathbf{c}) \right. \\
&\qquad\qquad\qquad\qquad\qquad\qquad\qquad \left. - \mathop{\mathbb{E}}_{\mathbf{c}'\sim P_t(\mathbf{C}'|\mathbf{X}=\mathbf{x})} P^h(Y \neq K + 1 \mid \mathbf{C}' = \mathbf{c}') \right] \\
&:= \pi_{t\wedge s}(\mathbf{X}, Y) + \pi_{t\setminus s}(\mathbf{X}, Y)
\end{aligned}
\tag{32}
$$

## B.3. Proof of Theorem 2

Given a domain $\mathcal{S}/\mathcal{T}$, an invariant representation inference model $G$, an intervention model $G'$, and a classifier $h$. For any probability distribution $\rho$ over $h$, we define the expected joint error $e_{\mathcal{S}/\mathcal{T}}(\mathbf{x}, \rho)$ and the expected disagreement $d_{\mathcal{S}/\mathcal{T}}(\mathbf{x}, \rho)$ as

$$
e_{\mathcal{T}}(\mathbf{x}) := \mathop{\mathbb{E}}_{h_1\sim\rho} \mathop{\mathbb{E}}_{h_2\sim\rho} \left( \mathop{\mathbb{E}}_{(\mathbf{x},y)\sim\mathcal{T}} \mathrm{I}\left(h_1(\mathbf{x}) \neq y\right) \mathrm{I}\left(h_2(\mathbf{x}) \neq y\right) \right)
\tag{33}
$$

$$
d_{\mathcal{T}_{\mathbf{X}}}(\mathbf{x}) := \mathop{\mathbb{E}}_{h_1\sim\rho} \mathop{\mathbb{E}}_{h_2\sim\rho} \left( \mathop{\mathbb{E}}_{(\mathbf{x},y)\sim\mathcal{T}_{\mathbf{X}}} \mathrm{I}\left(h_1(\mathbf{x}) \neq h_2(\mathbf{x})\right) \right)
\tag{34}
$$

Then we can decompose the susceptibility risk $\mathrm{SR}_t(\mathbf{c}, \mathbf{c}')$ as

$$
\begin{aligned}
\mathrm{SR}_{\mathcal{T}}(\mathbf{c}, \mathbf{c}') &= O_t(\mathbf{c}) - I_t(\mathbf{c}') \\
&= \frac{1}{2} \mathop{\mathbb{E}}_{(\mathbf{x},y)\sim\mathcal{T}_{\mathbf{X}}} \mathop{\mathbb{E}}_{\mathbf{c}\sim P_t^G(\mathbf{C}|\mathbf{X}=\mathbf{x})} \left[ \mathop{\mathbb{E}}_{h_1\sim\rho} \mathop{\mathbb{E}}_{h_2\sim\rho} \mathrm{I}[h_1(\mathbf{c}) \neq y] + \mathrm{I}[h_2(\mathbf{c}') \neq y] \right] \\
&\quad - \frac{1}{2} \mathop{\mathbb{E}}_{(\mathbf{x},y)\sim\mathcal{T}_{\mathbf{X}}} \mathop{\mathbb{E}}_{\mathbf{c}'\sim P_t^{G'}(\mathbf{C}'|\mathbf{X}=\mathbf{x})} \left[ \mathop{\mathbb{E}}_{h_1\sim\rho} \mathop{\mathbb{E}}_{h_2\sim\rho} \mathrm{I}[h_1(\mathbf{c}') \neq y] + \mathrm{I}[h_2(\mathbf{c}') \neq y] \right] \\
&= \mathop{\mathbb{E}}_{(\mathbf{x},y)\sim\mathcal{T}_{\mathbf{X}}} \mathop{\mathbb{E}}_{\mathbf{c}\sim P_t^G(\mathbf{C}|\mathbf{X}=\mathbf{x})} \left[ \mathop{\mathbb{E}}_{h_1\sim\rho} \mathop{\mathbb{E}}_{h_2\sim\rho} \frac{\mathrm{I}[h_1(\mathbf{c}) \neq h_2(\mathbf{c})] + 2\mathrm{I}[h_1(\mathbf{c}) \neq y \wedge h_2(\mathbf{c}) \neq y]}{2} \right] \\
&\quad - \mathop{\mathbb{E}}_{(\mathbf{x},y)\sim\mathcal{T}_{\mathbf{X}}} \mathop{\mathbb{E}}_{\mathbf{c}'\sim P_t^G(\mathbf{C}'|\mathbf{X}=\mathbf{x})} \left[ \mathop{\mathbb{E}}_{h_1\sim\rho} \mathop{\mathbb{E}}_{h_2\sim\rho} \frac{\mathrm{I}[h_1(\mathbf{c}') \neq h_2(\mathbf{c}')] + 2\mathrm{I}[h_1(\mathbf{x}) \neq y \wedge h_2(\mathbf{x}) \neq y]}{2} \right] \\
&= \frac{1}{2}(d_{\mathcal{T}_{\mathbf{X}}}(\mathbf{c}) - d_{\mathcal{T}_{\mathbf{X}}}(\mathbf{c}')) + e_{\mathcal{T}}(\mathbf{c}) - e_{\mathcal{T}}(\mathbf{c}').
\end{aligned}
\tag{35}
$$

We refer to the technicals in (Germain et al., 2016) by using $\beta$ divergence, which is formalized as $\beta_q(\mathcal{T}\|\mathcal{S}) = \left[ \mathop{\mathbb{E}}_{(\mathbf{x},y)\sim\mathcal{S}} \left( \frac{\mathcal{T}(\mathbf{x},y)}{\mathcal{S}(\mathbf{x},y)} \right)^q \right]^{\frac{1}{q}}$, and Hölder's Inequality, that is $\int |fg| \, d\mu \leq \left( \int |f|^q \, d\mu \right)^{\frac{1}{q}} \left( \int |g|^p \, d\mu \right)^{\frac{1}{p}}$, where $g = 1, q \rightarrow$

$+\infty, p \to 1$ in this paper. First, we define $r = \mathbb{E}_{(\mathbf{x},y)\sim\mathcal{T}}[I(\mathbf{x},y) \notin \text{supp}(\mathcal{S})]$, then using Hölder's Inequality, we have

$$
\begin{aligned}
e_{\mathcal{T}}(\mathbf{c}) &= \mathop{\mathbb{E}}_{(\mathbf{x},y)\sim\mathcal{T}_{\mathbf{x}}} \left[ \mathop{\mathbb{E}}_{\mathbf{c}\sim P_t^G(\mathbf{C}|\mathbf{X}=\mathbf{x})} \mathop{\mathbb{E}}_{h_1\sim\rho} \mathop{\mathbb{E}}_{h_2\sim\rho} \mathrm{I}(h_1(\mathbf{c}) \neq y)\,\mathrm{I}(h_2(\mathbf{c}) \neq y) \right] \\
&= \mathop{\mathbb{E}}_{(\mathbf{x},y)\sim\mathcal{S}} \left( \frac{\mathcal{T}(\mathbf{x},y)}{\mathcal{S}(\mathbf{x},y)} \right) \left[ \mathop{\mathbb{E}}_{\mathbf{c}\sim P_t^G(\mathbf{C}|\mathbf{X}=\mathbf{x})} \mathop{\mathbb{E}}_{h_1\sim\rho} \mathop{\mathbb{E}}_{h_2\sim\rho} \mathrm{I}(h_1(\mathbf{c}) \neq y)\,\mathrm{I}(h_2(\mathbf{c}) \neq y) \right] \\
&\quad + \mathbb{E}_{(\mathbf{x},y)\sim\mathcal{T}}[I(\mathbf{x},y) \notin \text{supp}(\mathcal{S})] \left[ \mathop{\mathbb{E}}_{\mathbf{c}\sim P_t^G(\mathbf{C}|\mathbf{X}=\mathbf{x})} \mathop{\mathbb{E}}_{h_1\sim\rho} \mathop{\mathbb{E}}_{h_2\sim\rho} \mathrm{I}(h_1(\mathbf{c}) \neq y)\,\mathrm{I}(h_2(\mathbf{c}) \neq y) \right] \\
&\leq \lim_{q\to\infty} \left( \beta_q(\mathcal{T}\|\mathcal{S}) e_{\mathcal{S}}(\mathbf{c}) \right) + r e_{\mathcal{T}\backslash\mathcal{S}}(\mathbf{c}),
\end{aligned}
\tag{36}
$$

$$
e_{\mathcal{T}}(\mathbf{c}') \leq \lim_{q\to\infty} \beta_q(\mathcal{T}\|\mathcal{S}) e_{\mathcal{S}}(\mathbf{c}') + r e_{t\backslash s}(\mathbf{c}')
\tag{37}
$$

Rewrite $r(e_{t\backslash s}(\mathbf{c}) - e_{t\backslash s}(\mathbf{c}'))$ as $\pi_{t\backslash s}(\mathbf{X}, Y)$. Then we have

$$
\begin{aligned}
\text{SR}_{\mathcal{T}}(\mathbf{c}, \mathbf{c}') &< \frac{1}{2}(d_{\mathcal{T}_{\mathbf{x}}}(\mathbf{c}) - d_{\mathcal{T}_{\mathbf{x}}}(\mathbf{c}')) \\
&\quad + \lim_{q\to\infty} \beta_q(\mathcal{T}\|\mathcal{S})(e_{\mathcal{S}}(\mathbf{c}) - e_{\mathcal{S}}(\mathbf{c}')) \\
&\quad + \pi_{t\backslash s}(\mathbf{X}, Y)
\end{aligned}
\tag{38}
$$

Then we can get the result of Theorem 2.

### B.4. Proof of Corollary 1

First, we introduce the PAC-Bayesian theorem (Catoni, 2007), which gives the usual bound on the empirical risk.

**Lemma 2.** *For domain $\mathcal{S}/\mathcal{T}$, any set of voters $\mathcal{H}$, any prior $\gamma$ over $\mathcal{H}$, any risk $\mathrm{R}: \mathcal{H} \to [0,1]$, any real number $c > 0$, with a probability at least $1 - \delta$ over the choice of $\{(\mathbf{x}_i, y_i)\}_{i=1}^{n} \sim \mathcal{T}$, we have for all $\rho$ on $\mathcal{H}$:*

$$
\begin{aligned}
&\mathop{\mathbb{E}}_{(\mathbf{x},y)\sim\mathcal{S}/\mathcal{T}} \mathop{\mathbb{E}}_{h\sim\rho} \mathrm{R}(\rho, \mathbf{x}, y) \\
&\leq \frac{c}{1 - e^{-c}} \left[ \frac{1}{n} \sum_{i=1}^{n} \mathop{\mathbb{E}}_{h\sim\rho} \mathrm{R}(\rho, \mathbf{x}_i, y_i) + \frac{\mathrm{KL}(\rho\|\gamma) + \ln\frac{1}{\delta}}{n \times c} \right].
\end{aligned}
\tag{39}
$$

Following the same setting of lemma 2, we have: with prior $\gamma_{\mathbf{C}}$, there exists a probability of at least $1 - \delta$ over $\mathcal{D}_t = \{(\mathbf{x}_i)\}_{i=1}^{n_t} \sim \mathcal{T}_{\mathbf{X}}$,

$$
\begin{aligned}
\forall \rho \text{ on } \mathcal{H}, d_{\mathcal{T}_{\mathbf{x}}}(\rho, \mathbf{c}) &\leq \frac{c}{1 - e^{-c}} \left[ d_{\mathcal{D}_t}(\rho, \mathbf{c}) + \frac{2\mathrm{KL}(\rho\|\gamma_{\mathbf{C}}) + \ln\frac{1}{\delta}}{n_t \times c} \right], \\
e_{\mathcal{S}}(\rho, \mathbf{c}) &\leq \frac{c}{1 - e^{-c}} \left[ e_{\mathcal{D}_s}(\rho, \mathbf{c}) + \frac{2\mathrm{KL}(\rho\|\gamma_{\mathbf{C}}) + \ln\frac{1}{\delta}}{n_s \times c} \right],
\end{aligned}
\tag{40}
$$

with prior $\gamma_{\mathbf{C}'}$, there exists a probability of at least $1 - \delta$ over $\mathcal{D}_s = \{(\mathbf{x}_i, y_i)\}_{i=1}^{n_s} \sim \mathcal{S}$,

$$
\begin{aligned}
\forall \rho \text{ on } \mathcal{H}, d_{\mathcal{T}_{\mathbf{x}}}(\rho, \mathbf{c}') &\leq \frac{c}{1 - e^{-c}} \left[ d_{\mathcal{D}_t}(\rho, \mathbf{c}') + \frac{2\mathrm{KL}(\rho\|\gamma_{\mathbf{C}'}) + \ln\frac{1}{\delta}}{n_t \times c} \right], \\
e_{\mathcal{S}}(\rho, \mathbf{c}') &\leq \frac{c}{1 - e^{-c}} \left[ e_{\mathcal{D}_s}(\rho, \mathbf{c}') + \frac{2\mathrm{KL}(\rho\|\gamma_{\mathbf{C}'}) + \ln\frac{1}{\delta}}{n_s \times c} \right],
\end{aligned}
\tag{41}
$$

Then, we obtain the following generalization bound defined with respect to the empirical estimates of the target disagreement and the source joint error: with a probability at least $1 - \delta$ over $\mathcal{D}_s = \{(\mathbf{x}_i, y_i)\}_{i=1}^{n_s} \sim \mathcal{T}_{\mathbf{X}}$ and $\mathcal{D}_t = \{(\mathbf{x}_i)\}_{i=1}^{n_t} \sim \mathcal{T}_{\mathbf{X}}$, let

$b' = \frac{b}{1-e^{-b}}\beta_\infty(\mathcal{T}\|\mathcal{S})$, and $c' = \frac{c}{1-e^{-c}}$, $\forall \rho$ on $\mathcal{H}$,

$$
\begin{aligned}
\mathrm{SR}_\mathcal{T}(\mathbf{c}, \mathbf{c}') \leq &\; \frac{c'}{2}(d_{\mathcal{D}_t}(\mathbf{c}) - d_{\mathcal{D}_t}(\mathbf{c}')) + b'(e_{\mathcal{D}_s}(\mathbf{c}) - e_{\mathcal{D}_s}(\mathbf{c}')) \\
&+ R_\mathcal{S}^\mathcal{U}(\mathbf{X}, Y) + \left(\frac{c'}{m_t \times c} + \frac{b'}{m_s \times b}\right)\left(2\mathrm{KL}(\rho\|\gamma_\mathbf{C}) + \ln\frac{2}{\delta}\right) \\
&- \left(\frac{c'}{m_t \times c} + \frac{b'}{m_s \times b}\right)\left(2\mathrm{KL}(\rho\|\gamma_{\mathbf{C}'}) + \ln\frac{2}{\delta}\right) \\
= &\; \frac{c'}{2}(d_{\mathcal{D}_t}(\mathbf{c}) - d_{\mathcal{D}_t}(\mathbf{c}')) + b'(e_{\mathcal{D}_s}(\mathbf{c}) - e_{\mathcal{D}_s}(\mathbf{c}')) + R_\mathcal{S}^\mathcal{U}(\mathbf{X}, Y) \\
&+ 2\left(\frac{c'}{m_t \times c} + \frac{b'}{m_s \times b}\right)(\mathrm{KL}(\rho\|\gamma_\mathbf{C}) + \mathrm{KL}(\gamma_{\mathbf{C}'}\|\rho))
\end{aligned}
\tag{42}
$$

Therefore, we can narrow the error between $\mathrm{SR}_\mathcal{T}(\mathbf{c})$ and $\mathrm{SR}_{\mathcal{D}_t}(\mathbf{c})$ by minimizing $(\mathrm{KL}(\rho\|\gamma_\mathbf{C}) + \mathrm{KL}(\gamma_{\mathbf{C}'}\|\rho))$. Then with a non-informative prior $\gamma := \mathcal{N}(\mu_\gamma, \sigma_\gamma^2)$, the produced Gaussian $\mathcal{N}(\mu_\mathbf{c}, \sigma_\mathbf{c}^2)$, we have

$$
\begin{aligned}
&\min_{\mu_\mathbf{c}, \sigma_\mathbf{c}}\left(\lim_{\sigma_\gamma \to \infty} \mathrm{KL}\left(\mathcal{N}\left(\mu_\mathbf{c}, \sigma_\mathbf{c}^2\right)\|\mathcal{N}\left(\mu_\gamma, \sigma_\gamma^2\right)\right)\right) \\
&\Rightarrow \min_{\mu_\mathbf{c}, \sigma_\mathbf{c}}\left(\lim_{\sigma_\gamma \to \infty}\left(\log\frac{\sigma_\gamma}{\sigma_\mathbf{c}} + \frac{\sigma_\mathbf{c}^2 + (\mu_\mathbf{c} - \mu_\gamma)^2}{2\sigma_\gamma^2} - \frac{1}{2}\right)\right) \\
&\Rightarrow \min_{\sigma_\mathbf{c}}\left(\lim_{\sigma_\gamma \to \infty}\left(\log\frac{\sigma_\gamma}{\sigma_1}\right)\right) \Rightarrow \min_{\sigma_\mathbf{c}}\left(-\log\sigma_\mathbf{c}\right).
\end{aligned}
\tag{43}
$$

Finally, we can minimize $\mathrm{SR}_\mathcal{T}(\mathbf{c})$ by adding Eq. 14, which is the result of Corollary 1.

## B.5. Proof of Proposition 2

According to the definition of the KL divergence and the Mutual Information $I(\cdot; \cdot)$, we can rewrite $\mathrm{Pr}(Y = k|\mathbf{C}, \mathbf{E} = \mathrm{e}_{t/s}) = \mathrm{Pr}(Y = k \mid \mathbf{C})$ as:

$$
\begin{aligned}
&\mathrm{KL}(P(Y = k \mid \mathbf{C}, \mathbf{E})\|P(Y = k \mid \mathbf{C})) \\
&= P(Y = k \mid \mathbf{C}, \mathbf{E})\log\left(\frac{P(Y = k \mid \mathbf{C}, \mathbf{E})}{P(Y = k \mid \mathbf{C})}\right) = 0 \\
&= p(\mathbf{C}, \mathbf{E}, Y = k)\log\left(\frac{p(Y = k, \mathbf{E} \mid \mathbf{C})}{p(Y = k \mid \mathbf{C})p(\mathbf{E} \mid \mathbf{C})}\right) = I(Y = k; \mathrm{e} \mid \mathbf{C})
\end{aligned}
\tag{44}
$$

Based on $I(Y = k; \mathrm{e} \mid \mathbf{C}) = 0$, we have

$$
I(Y; \mathbf{C}|\mathrm{e}_t) = I(Y; \mathbf{C}) = I(Y; \mathbf{C}|\mathrm{e}_s)
\tag{45}
$$

Based on the transmissibility of $I(\cdot; \cdot)$, $I(Y; \mathbf{C}|\mathrm{e}_t) = I(Y; \mathbf{C}|\mathrm{e}_s)$ can be achieved by:

$$
\max I(\mathbf{C}, \mathrm{e}_s; \mathbf{C}, \mathrm{e}_t)
\tag{46}
$$

Finally, given the universal data generation causal model $M$, $M : \mathbf{C}, \mathbf{E} \to \mathbf{X}$, the causal representation model $G$, $G : \mathbf{X} \to \mathbf{C}$, we reach

$$
\begin{aligned}
&\max_G I(G(M(\mathbf{C}, \mathrm{e}_s)); G(M(\mathbf{C}, \mathrm{e}_t))) \\
&\Rightarrow \max_G I(G(\mathbf{x}_s); G(\mathbf{x}_t))
\end{aligned}
\tag{47}
$$

Then we can get the result of Proposition 2.

# C. Related Works & Our Innovations

In this section, we review the progress of OSDA prediction tasks, including:

First, adversarial learning-based algorithms. OSBP (Saito et al., 2018) first introduces the concept of domain adversarial learning into OSDA. Unlike UDA, OSBP sets a fixed threshold $t$ for separating unknown class samples and employs gradient reversal to achieve a min-max adversarial optimization between the feature extractor and classifier. PGL (Luo et al., 2020) proposed a Progressive Graph Learning framework based on OSBP, which integrates graph neural networks trained in context to mitigate potential conditional shifts. Recently, ANNA (Li et al., 2023) considered the unknown class information embedded in the background features of known class samples, then introduced causal unbiased learning and domain alignment on top of OSBP.

Second, score-based algorithms. Separate to Adapt (STA) (Liu et al., 2019) generates weights to reject target samples belonging to unknown classes while balancing their importance in feature distribution alignment. Inspired by STA's approach, (Shermin et al., 2020) introduced a weighting module based on the similarity between target samples and source domain classes to find an appropriate threshold for each sample. ROS (Bucci et al., 2020) utilizes rotational invariance to improve source domain models and proposes entropy- and confidence-based regularization scores to distinguish unknown class samples. UAN (You et al., 2019) proposed a transferability criterion to quantify sample-level transferability, aiming to discover the common label set and the label sets private to each domain. GATE (Chen et al., 2022) introduced a generic incremental classifier that adaptively learns the "unknown" threshold by minimizing open-set entropy.

Third, pseudo-label-based alignment strategies. OSLPP (Wang et al., 2024) calculates class means and uses them to assign pseudo-labels to target domain samples, while target samples that are "far from" all known class means are recognized as unknown classes. This approach suffers from two key limitations: first, the ambiguous definition of "far from"; second, its inability to identify unknown-class samples that are similar yet distinct from known classes. DCC proposes Domain Consensus Clustering for universal domain adaptation, using semantical and sample-level consensus to effectively separate and distinguish common classes from private ones. OVANet proposes a universal domain adaptation method that learns an open-set threshold from source data via one-vs-all classifiers and adapts it to the target domain by minimizing class entropy. UADAL addresses open-set domain adaptation with unknown-aware adversarial learning, aligning known classes while segregating unknowns in feature space. Recently, GLC (Qu et al., 2023) generated positive prototypes using the Top-$K$ most reliable samples for each class and then applied a clustering algorithm to the remaining samples to create specific negative prototypes. The class of a sample is determined by comparing its distance to both positive and negative prototypes. The computational complexity of this method is proportional to the number of known classes, which reduces learning efficiency on large datasets. Inspired by GLC, we propose a simpler and more efficient approach to pseudo-label generation, called the Virtual Multi-unknown-categories Prototype (VMP) pseudo-labeling strategy. The difference is that we exclude all known-class positive samples and cluster only the remaining negative samples (i.e., those not belonging to any known class). Besides, unlike one-vs-all strategies requiring $C$ clustering operations per epoch ($C$ = the number of known classes), COSDA achieves comparable performance with only one clustering per epoch.

Recent advances in Open Set Domain Adaptation (OSDA) have primarily focused on three key research directions: (1) enhancing source domain knowledge acquisition, (2) establishing effective boundaries for unknown class detection, and (3) aligning known class distributions across domains. Our proposed COSDA framework makes significant contributions to all three areas. For source model learning, we introduce a novel causal susceptibility risk that guides the model to capture essential causal features, with theoretical analysis providing learning upper bounds for the OSDA setting. For unknown class separation, we propose the Virtual Multi-Class Prototype (VMCP) method, which generates multiple virtual unknown class prototypes through K-means clustering on global target samples while filtering potential known class prototypes, enabling robust pseudo-label assignment via similarity matching. For cross-domain alignment, we develop a class-level contrastive learning approach that leverages pseudo-labels to effectively align feature representations of known categories between source and target domains.

# D. Details for Experiments.

## D.1. Benchmark Dataset Settings.

Extensive experiments are conducted on five benchmarks following the standard setting (Qu et al., 2023; Bucci et al., 2020): 1) *Office-Home* (Venkateswara et al., 2017), a dataset across four distinct domains: Art (Ar), Clipart (Cl), Product (Pr), and Real World (Rw) with the first 25 categories as the known, while the subsequent 40 classes as the unknown; 2)

*Office-31* (Wang et al., 2019), another dataset spanning three domains: Amazon (A), Dslr (D), and Webcam (W), with the first 10 classes as the known and the last 11 classes as the unknown; 3) *Image-CLEF* (Li et al., 2023), a database with four domains and 12 shared common classes. The first 6 classes are utilized as the known, and the rest as the unknown; 4) *VisDA* (Li et al., 2021), a synthetic-to-real (S2R) dataset with 12 classes, with first 6 classes as the known and the rest classes as the unknown; 5) DomainNet (Wen & Brbic, 2024) is the largest dataset, including 345 classes and six domains. Following the previous work (Wen & Brbic, 2024), we use three domains: Painting (P), Real (R), and Sketch (S).

| Dataset | #domains | #K | #U | #images |
|---------|----------|-----|-----|---------|
| *Office-Home* | 4 | 25 | 40 | 15,500 |
| *Office-31* | 3 | 10 | 11 | 4,652 |
| *Image-CLEF* | 4 | 6 | 6 | 600 |
| *VisDA* | 2 | 6 | 6 | 207,785 |
| *DomainNet* | 3 | 173 | 172 | 341,472 |

*Table 5.* Dataset details for open-set domain adaptation.

## D.2. Baselines

**Baselines.** The baselines include i) adversarial-learning methods, i.e., **OSBP** (Saito et al., 2018), **PGL** (Luo et al., 2020), **ANNA** (Li et al., 2023), ii) score-based methods, i.e., **STA** (Liu et al., 2019), **ROS** (Bucci et al., 2020), **UAN** (You et al., 2019), **GATE** (Chen et al., 2022), iii) pseudo-label-based alignment methods, i.e., **DADO** (Fang et al., 2020), **DCC** (Li et al., 2021), **OVANet** (Saito, 2021). **OSLPP** (Wang et al., 2024), **GLC** (Qu et al., 2023), **UPUK** (Wan et al., 2024), **USDAP** (Shao et al., 2024).

## D.3. Evaluation Metrics

Following the main OSDA stream (Bucci et al., 2020; Qu et al., 2023; Saito et al., 2018; Li et al., 2023), we utilize four widely used measures, i.e., accuracy of the unknown class ($UNK$), normalized accuracy for the known classes only ($OS^*$), normalized accuracy for all classes($OS$), and harmonic mean accuracy ($HOS$).

$$\text{UNK} = \frac{1}{N^{K+1}} \mathop{\mathbb{E}}_{(x,y)\sim\mathcal{D}_t^{K+1}} \mathbb{I}(h(x) = y), \tag{48}$$

$$\text{OS*} = \frac{1}{K} \sum_{k=1}^{K} \frac{1}{N^k} \mathop{\mathbb{E}}_{(x,y)\sim\mathcal{D}_t^{k+1}} \mathbb{I}(h(x) = y), \tag{49}$$

$$\text{OS} = \frac{1}{K+1} \sum_{k=1}^{K+1} \frac{1}{N^k} \mathop{\mathbb{E}}_{(x,y)\sim\mathcal{D}_t^{k+1}} \mathbb{I}(h(x) = y), \tag{50}$$

$$\text{HOS} = \frac{2 \times \text{OS*} \times \text{UNK}}{\text{OS*} + \text{UNK}}. \tag{51}$$

with $D_t^i$ being the set of target samples in the $i$-th class, and $h()$ the classifier. *HOS* is regarded as the most equitable evaluation criterion. It strikes a balance between assessing the performance of the methods on both known and unknown class samples.

## D.4. Additional Benchmark Experiments Results

Table 6 shows the comparison results of *Office-Home*.

## D.5. Details for Synthetic Numerical Experiment

In this section, we conduct two synthetic experiments to evaluate the effectiveness of Susceptibility risk.

**Synthetic Numerical Experiment**

*Table 6.* Comparison results (%) of *Office-Home*. (Best in **bold** and second best in underline)

| Method | Average OS* | UNK | HOS | Ar→Cl OS* | UNK | HOS | Ar→Pr OS* | UNK | HOS | Ar→Rw OS* | UNK | HOS | Cl→Ar OS* | UNK | HOS | Cl→Pr OS* | UNK | HOS |
|---|---|---|---|---|---|---|---|---|---|---|---|---|---|---|---|---|---|---|
| OSBP (Saito et al., 2018) | 64.1 | 66.3 | 64.7 | 50.2 | 61.1 | 55.1 | 71.8 | 59.8 | 65.2 | 79.3 | 67.5 | 72.9 | 59.4 | 70.3 | 64.3 | 67.0 | 62.7 | 64.7 |
| STAsum (Liu et al., 2019) | 63.4 | 62.6 | 61.9 | 50.8 | 63.4 | 56.3 | 68.7 | 59.7 | 63.7 | 81.1 | 50.5 | 62.1 | 53.0 | 63.9 | 57.9 | 61.4 | 63.5 | 62.5 |
| STAmax (Liu et al., 2019) | 61.8 | 63.3 | 61.6 | 46.0 | 72.3 | 55.8 | 68.0 | 48.4 | 54.0 | 78.6 | 60.4 | 68.3 | 51.4 | 65.0 | 57.4 | 61.8 | 59.1 | 60.4 |
| UAN (You et al., 2019) | 75.2 | 0.0 | 0.1 | 62.4 | 0.0 | 0.0 | 81.1 | 0.0 | 0.0 | 88.2 | 0.1 | 0.2 | 70.5 | 0.0 | 0.0 | 74.0 | 0.1 | 0.2 |
| PGL (Luo et al., 2020) | 76.1 | 25.0 | 35.2 | 63.3 | 19.1 | 29.3 | 78.9 | 32.1 | 45.6 | 87.7 | 40.9 | 55.8 | 85.9 | 5.3 | 10.0 | 73.9 | 24.5 | 36.8 |
| ROS (Bucci et al., 2020) | 61.6 | 72.4 | 66.2 | 50.6 | 74.1 | 60.1 | 68.4 | 70.3 | 69.3 | 75.8 | 77.2 | 76.5 | 53.6 | 65.5 | 58.9 | 59.8 | 71.6 | 65.2 |
| DAOD (Fang et al., 2020) | 69.6 | 50.2 | 57.6 | 72.6 | 51.8 | 60.5 | 55.3 | 57.9 | 56.6 | 78.2 | 62.6 | 69.5 | 59.1 | 61.7 | 60.4 | 70.8 | 52.6 | 60.4 |
| DCC (Li et al., 2021) | - | - | 61.7 | - | - | 56.1 | - | - | 67.5 | - | - | 66.7 | - | - | 49.6 | - | - | 66.5 |
| OVANet (Saito, 2021) | - | - | 64.0 | - | - | 58.6 | - | - | 66.3 | - | - | 69.9 | - | - | 62.0 | - | - | 65.2 |
| GATE (Chen et al., 2022) | - | - | 69.0 | - | - | 63.8 | - | - | 70.5 | - | - | 75.8 | - | - | 66.4 | - | - | 67.9 |
| ANNA (Li et al., 2023) | 65.6 | 76.7 | 70.7 | 61.4 | 78.7 | 69.0 | 68.3 | 79.9 | 73.7 | 74.1 | 79.7 | 76.8 | 58.0 | 73.1 | 64.7 | 64.2 | 73.6 | 68.6 |
| GLC (Qu et al., 2023) | - | - | 69.8 | - | - | 65.3 | - | - | 74.2 | - | - | 79.0 | - | - | 60.4 | - | - | 71.6 |
| OSLPP (Wang et al., 2024) | 63.8 | 71.7 | 67.0 | 55.9 | 67.1 | 61.0 | 72.5 | 73.1 | 72.8 | 80.1 | 69.4 | 74.3 | 49.6 | 79.0 | 60.9 | 61.6 | 73.3 | 66.9 |
| UPUK (Wan et al., 2024) | 61.6 | 79.7 | 69.2 | 49.0 | 64.8 | 55.8 | 68.1 | 88.0 | 76.7 | 71.5 | 86.6 | 78.4 | 61.3 | 72.6 | 66.4 | 66.5 | 81.6 | 73.1 |
| USDAP (Shao et al., 2024) | - | - | 69.6 | - | - | 66.7 | - | - | 75.3 | - | - | 76.2 | - | - | 60.7 | - | - | 70.0 |
| GLC++ (Qu et al., 2024) | - | - | 70.2 | - | - | 65.4 | - | - | 73.8 | - | - | 78.0 | - | - | 61.5 | - | - | 71.9 |
| **Ours** | 69.4 | 74.4 | 71.7 | 70.0 | 71.0 | 70.5 | 73.9 | 76.0 | 74.9 | 80.3 | 79.5 | 79.9 | 57.9 | 74.0 | 65.0 | 68.8 | 76.0 | 72.2 |

| Method | Cl→Rw OS* | UNK | HOS | Pr→Ar OS* | UNK | HOS | Pr→Cl OS* | UNK | HOS | Pr→Rw OS* | UNK | HOS | Rw→Ar OS* | UNK | HOS | Rw→Cl OS* | UNK | HOS | Rw→Pr OS* | UNK | HOS |
|---|---|---|---|---|---|---|---|---|---|---|---|---|---|---|---|---|---|---|---|---|---|
| OSBP | 72.0 | 69.2 | 70.6 | 59.1 | 68.1 | 63.2 | 44.5 | 66.3 | 53.2 | 76.2 | 71.7 | 73.9 | 66.1 | 67.3 | 66.7 | 48.0 | 63.0 | 54.5 | 76.3 | 68.6 | 72.3 |
| STAsum | 69.8 | 63.2 | 66.3 | 55.4 | 73.7 | 63.1 | 44.7 | 71.5 | 55.0 | 78.1 | 63.3 | 69.7 | 67.9 | 62.3 | 65.0 | 51.4 | 57.9 | 54.2 | 77.9 | 58.0 | 66.4 |
| STAmax | 67.0 | 66.7 | 66.8 | 54.2 | 72.4 | 61.9 | 44.2 | 67.1 | 53.2 | 76.2 | 64.3 | 69.5 | 67.5 | 66.7 | 67.1 | 49.9 | 61.1 | 54.5 | 77.1 | 55.4 | 64.5 |
| UAN | 80.6 | 0.1 | 0.2 | 73.7 | 0.0 | 0.0 | 59.1 | 0.0 | 0.0 | 84.0 | 0.1 | 0.2 | 77.5 | 0.1 | 0.2 | 66.2 | 0.0 | 0.0 | 85.0 | 0.1 | 0.1 |
| PGL | 70.2 | 33.8 | 45.6 | 73.7 | 34.7 | 47.2 | 59.2 | 38.4 | 46.6 | 84.8 | 27.6 | 41.6 | 81.5 | 6.1 | 11.4 | 68.8 | 0.0 | 0.0 | 84.8 | 38.0 | 52.5 |
| ROS | 65.3 | 72.2 | 68.6 | 57.3 | 64.3 | 60.6 | 46.5 | 71.2 | 56.3 | 70.8 | 78.4 | 74.4 | 67.0 | 70.8 | 68.8 | 51.5 | 73.0 | 60.4 | 72.0 | 80.0 | 75.7 |
| DADO | 77.8 | 57.0 | 65.8 | 71.3 | 50.5 | 59.1 | 58.4 | 42.8 | 49.4 | 81.8 | 50.6 | 62.5 | 66.7 | 43.3 | 52.5 | 60.0 | 36.6 | 45.5 | 84.1 | 34.7 | 49.1 |
| DCC | - | - | 64.0 | - | - | 55.8 | - | - | 53.0 | - | - | 70.5 | - | - | 61.6 | - | - | 57.2 | - | - | 71.9 |
| OVANet | - | - | 68.6 | - | - | 59.8 | - | - | 53.4 | - | - | 69.3 | - | - | 68.7 | - | - | 59.6 | - | - | 66.7 |
| GATE | - | - | 71.7 | - | - | 67.3 | - | - | 61.5 | - | - | 76.0 | - | - | 70.4 | - | - | 61.8 | - | - | 75.1 |
| ANNA | 66.9 | 80.2 | 73.0 | 63.0 | 70.3 | 66.5 | 54.6 | 74.8 | 63.1 | 74.3 | 78.9 | 76.6 | 66.1 | 77.3 | 71.3 | 59.7 | 73.1 | 65.7 | 76.4 | 81.0 | 78.7 |
| GLC | - | - | 74.7 | - | - | 63.7 | - | - | 63.2 | - | - | 75.8 | - | - | 67.1 | - | - | 64.3 | - | - | 77.8 |
| OSLPP | 67.2 | 73.9 | 70.4 | 54.6 | 76.2 | 63.6 | 53.1 | 67.1 | 59.3 | 77.0 | 71.2 | 74.0 | 60.8 | 75.0 | 67.2 | 54.4 | 64.3 | 59.0 | 78.4 | 70.8 | 74.4 |
| UPU | 71.6 | 84.8 | 77.6 | 55.9 | 85.6 | 67.6 | 45.4 | 70.2 | 55.1 | 73.9 | 83.9 | 78.6 | 56.7 | 84.1 | 67.8 | 49.3 | 74.6 | 59.4 | 69.5 | 80.0 | 74.4 |
| USDAP | - | - | 72.6 | - | - | 64.5 | - | - | 64.6 | - | - | 76.5 | - | - | 65.5 | - | - | 64.3 | - | - | 78.0 |
| GLC++ | - | - | 74.7 | - | - | 64.2 | - | - | 65.3 | - | - | 76.0 | - | - | 67.7 | - | - | 66.0 | - | - | 77.8 |
| **Ours** | 76.4 | 75.8 | 76.1 | 62.1 | 74.4 | 67.8 | 58.5 | 70.0 | 63.7 | 73.3 | 79.5 | 76.3 | 66.9 | 74.0 | 70.3 | 63.4 | 64.9 | 64.1 | 81.1 | 77.3 | 79.1 |

The effectiveness of susceptibility risk is demonstrated by examining whether it can learn the essential causal relationships. To this end, in the first synthetic experiment, we designed a synthetic data generator based on Fig. 8 and created a sample set $\{\mathbf{X}, Y\}$, where $X$ is generated from three causal factors $\mathbf{C}$ and a spurious association factor $\mathbf{S}$. The generation process of $\mathbf{X}$ is as follows:

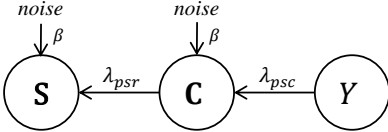

*Figure 8.* Causal graph for Synthetic Data Generation: $Y \perp S \mid C$

$$
\begin{aligned}
C_{psc} &= Y \oplus \mathcal{B}(\lambda_{psc}) + \beta \mathcal{N}(0, 1), \\
C_{mean} &= \text{Mean}(\{C_{psc}\}), \\
S &= \lambda_{psr} C_{mean} * \mathbf{1}_d + \beta \mathcal{N}(0, 1), \\
X &= \text{MLP}(\{\{C_{psc}\}, S\}).
\end{aligned}
\tag{52}
$$

In Eq. (52), to test whether susceptibility risk can focus on more essential causal information, we use a parameter $\lambda_{\text{psc}}$ to distinguish the importance of causal factors. When $\lambda_{\text{psc}}$ is closer to 0 or 1, the causal association is stronger to Y. Conversely, the closer $\lambda_{\text{psc}}$ is to 0.5, the weaker the causal association is. $\beta$ represents the noise intensity in the data. We set $\lambda_{\text{psc}} = \{0.15, 0.25, 0.35\}$ and $\beta = \{0.1, 0.4, 0.7, 1.1\}$. Besides, $\lambda_{psr}$ denotes the spurious degree. When $\lambda_{psr}$ gets higher, the spurious correlation is stronger in data $X$. We set $d = 64$ and $\lambda_{psr} = \{0.1, 0.4, 0.7\}$. Then we develop a non-linear function to generate $\mathbf{X}$ from $[C_{0.15}, C_{0.25}, C_{0.35}, S]$. We use distance correlation to evaluate the correlation between the

learned representation and these factors. A higher distance correlation indicates a better representation of factors. We provided an ablation study to demonstrate the efficiency of susceptibility risk by comparing it with vanilla empirical risk.

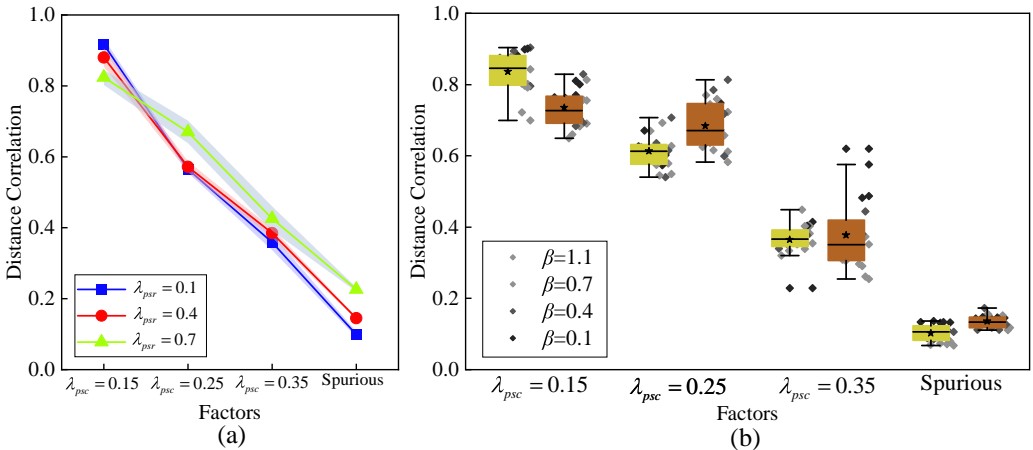

*Figure 9.* (a) Results of different spurious degree, and (b) Results of Suscepetibility risk ( yellow ) and Vanilla risk ( orange )

In Fig. 9(a), we varied the level of spuriousness and computed the distance correlation between the learned representations and four factors. It can be observed that susceptibility risk achieved a higher distance correlation with critical causal factors, while the distance correlation with spurious factors showed only a slight increase as spuriousness increased. This demonstrates that our evaluator consistently focuses on critical causal factors under varying spurious levels. In Fig. 9(b), we generated data with different levels of perturbation by varying $\beta$ and compared the performance of the Susceptibility risk evaluator with that of a standard empirical risk evaluator. For critical causal factors ($\lambda_{psc} = 0.15$), our method achieved a higher distance correlation than the standard risk. For factors with lower causal relevance ($\lambda_{psc} = 0.25$, $\lambda_{psc} = 0.35$), our method comparatively avoided overemphasis. Furthermore, for spurious factors, our method yielded a lower distance correlation compared to the standard risk. These results verified that by incorporating susceptibility risk evaluation, models can reduce the learning of spurious correlations and maintain a focus on critical causal factors.

**Action Recognition in Appliance Disassembly under OSDA.** To evaluate the practical applicability of the proposed method, we selected appliance disassembly action recognition as a concrete problem. Disassembly actions are complex, with similar backgrounds in images, and involve a significant number of unknown actions and unknown parts, which severely degrade the performance of traditional supervised learning models. The open-set domain adaptation setup used in this paper aligns with the challenges faced in action recognition during appliance disassembly at the test stage. Thus, we constructed a custom **Image** dataset for **A**ppliance **D**isassembly **A**ction **R**ecognition, **Image-ADAR**, for this task and compared the performance of COSDA with several baseline methods.

The dataset was collected in a disassembly factory, where each collection environment represents a distinct domain, labeled as domains A, B, and C. After data processing, each domain consists of 7 classes, including five shared classes and two unknown classes. The source domain includes only the five shared classes, while the target domain encompasses all categories. The objects being disassembled are air conditioners, with the shared classes being disassembling the outer shell, disassembling the evaporator, disassembling the foam, and disassembling the circuit board. The two unknown classes are action unknown (leaving the workbench) and object unknown (refrigerator). Each class contains between 150 and 450 image samples. In line with standard experimental settings, the OS*, UNK, and HOS classification metrics were used, and the experimental parameters were consistent with the *Office-31* configuration. The results are presented in Tab. 8.

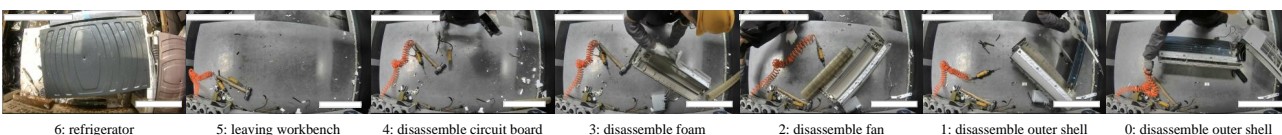

| 6: refrigerator | 5: leaving workbench | 4: disassemble circuit board | 3: disassemble foam | 2: disassemble fan | 1: disassemble outer shell | 0: disassemble outer shell |

*Figure 10.* Examples of each class in *Image-ADAR*, where label 5-6 are unknown categories, and label 0-4 are shared categories.

*Table 7.* Dataset details for *Image-ADAR*.

| Dataset | #domain name | #K | #U | #images |
|---------|--------------|----|----|---------|
| *Image-ADAR* | A | 5 | 2 | 1,573 |
|  | B | 5 | 2 | 1,951 |
|  | C | 5 | 2 | 1,823 |

*Table 8.* Comparison results (%) of *Image-ADAR*. (Best in **bold** and second best in underline)

| Method | A→B | | | A→C | | | B→A | | | B→C | | | C→A | | | C→B | | | Average | | |
|--------|-----|-----|-----|-----|-----|-----|-----|-----|-----|-----|-----|-----|-----|-----|-----|-----|-----|-----|-----|-----|-----|
|  | OS* | UNK | **HOS** | OS* | UNK | **HOS** | OS* | UNK | **HOS** | OS* | UNK | **HOS** | OS* | UNK | **HOS** | OS* | UNK | **HOS** | OS* | UNK | **HOS** |
| OSBP (Saito et al., 2018) | 56.1 | **79.8** | 65.9 | **74.9** | 73.5 | 74.2 | 73.1 | 62.5 | 67.4 | 52.4 | 37.6 | **43.8** | 44.2 | 14.2 | 21.5 | 29.9 | 41.5 | 34.7 | 55.1 | 51.5 | 51.2 |
| ANNA (Li et al., 2023) | 63.4 | 77.3 | 69.7 | 74.4 | 66.1 | 70.0 | **82.3** | **77.0** | **79.6** | **54.8** | 34.9 | 42.6 | 48.6 | 16.0 | 24.1 | 24.5 | 60.3 | 34.9 | 58.0 | 55.3 | 53.5 |
| **Ours** | **66.9** | 74.6 | **70.5** | 73.8 | **89.2** | **80.7** | 62.3 | 65.4 | 63.8 | 33.3 | **57.7** | 42.3 | **71.2** | 56.5 | 63.0 | 50.1 | 63.5 | 56.0 | 59.6 | 67.8 | 62.7 |

We compared our method with the classical baseline, OSBP, and the state-of-the-art OSDA method, ANNA. The results show that our method achieved the best performance of OS*, UNK, and HOS. Notably, COSDA achieves significant improvements over the baselines on the C → A task, with a 20.6% increase in OS*, a 40.5% increase in UNK, and a 38.9% increase in HOS, demonstrating the effectiveness of our proposed method.

### D.6. Qualitative Results

**Trade-off Analysis.** To investigate how the parameters in Eq. (25) affect the performance of COSDA, we define $\lambda_s = \{0.1, 0.2, 0.5, 0.8, 1.0\}$ and $\lambda_{exo} = \{0.1, 0.2, 0.5, 0.8, 1.0\}$. For each dataset, we select the final sub-task for parameter sensitivity analysis. As shown in Fig. 5, COSDA fluctuates between 93.8 and 94.8 on the P-I (*Image-CLEF*) task. On the Rw-Pr (*Office-Home*) task, COSDA shows a performance range of 77.8 to 78.8. This indicates that the proposed algorithm demonstrates strong parameter stability across both large and small datasets.

**Loss Convergence Analysis.** Fig. 11 depicts the loss convergence trends on the test sets of the *dslr → amazon* (*Office-31*) and *Art → Product* (*Office-Home*) tasks. Overall, the model achieves sufficient fitting within iterations without overfitting. Specifically, in both tasks, the sharp decline in $\mathcal{L}_t$ reflects a significant improvement in target domain classification accuracy driven by pseudo-labeling. Meanwhile, $\mathcal{L}_s$, representing source domain classification accuracy, exhibits a slight downward trend in the *dslr → amazon* task but remains relatively stable in the *Art → Product* task. Finally, $\mathcal{L}_{exo}$, which represents feature alignment, shows a gradual downward trend throughout the training process.

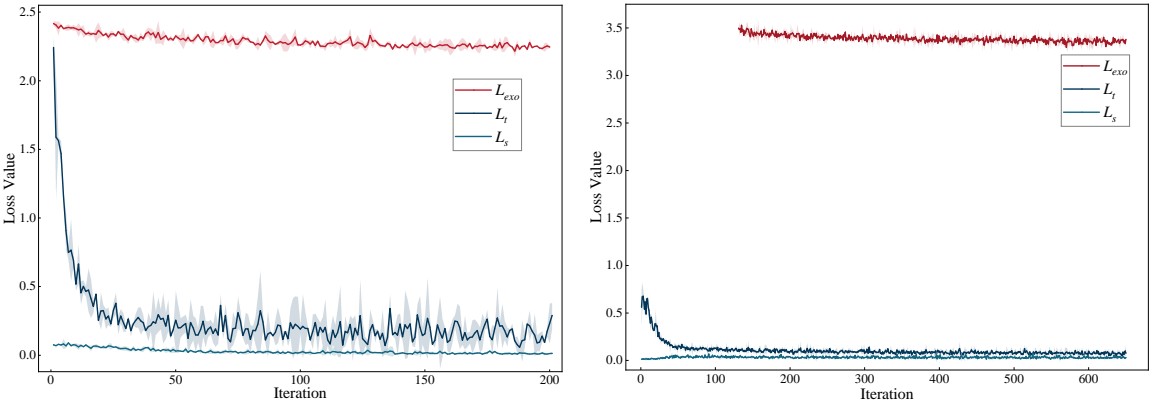

*Figure 11.* Loss convergence on *dslr → amazon* (left) and *Art → Product* (right)

