# OpenReview forum: "COSDA: Counterfactual-based Susceptibility Risk Framework for Open-Set Domain Adaptation"
_ICML.cc/2025/Conference — ICML 2025 poster_

### Official Review · Reviewer_rce5 · 2025-02-17

**Overall Recommendation:** 4

**Summary:**

This paper establishes a novel causal-inspired theoretical framework for Open-Set Domain Adaption by exploring the susceptibility between two visual samples. Based on the theoretical analysis, the authors propose three components: the SRE for estimating the causal relativity; the CFA module to facilitate cross-domain feature alignment; and the VMP strategy for pseudo labeling. The theoretical proof seems correct and novel, which fills up the gap of causal inference in the OSDA tasks.

**Claims And Evidence:**

What is the meaning of “c' is the specific implementation of c” in Definition 1 (LL94-95), it is a little bit confusing that I cannot relate it to any real examples. What is the connection between these variables to the source/target sample? Since this definition affects the following proofs and model design a lot, it should be more clearly defined. The authors could give a concrete example for intuitive understanding.

**Essential References Not Discussed:**

The second key contribution is a pseudo-labeling strategy using k-means for clustering in the target domain. A close relative idea has been proposed in prior work DCC[1] published in CVPR 2021, which is not discussed in the related work.
Since OSDA/UniDA has been studied for nearly 7 years, the discussion for previous works is not comprehensive. Several classical researches are missing, such as OVAnet [2] and UADAL [3].

[1] Li G, Kang G, Zhu Y, et al. Domain consensus clustering for universal domain adaptation[C]//Proceedings of the IEEE/CVF conference on computer vision and pattern recognition. 2021: 9757-9766.
[2] Saito K, Saenko K. Ovanet: One-vs-all network for universal domain adaptation[C]//Proceedings of the ieee/cvf international conference on computer vision. 2021: 9000-9009.
[3] Jang J H, Na B, Shin D H, et al. Unknown-aware domain adversarial learning for open-set domain adaptation[J]. Advances in Neural Information Processing Systems, 2022, 35: 16755-16767.

**Experimental Designs Or Analyses:**

The overall results in the experiment sections are comprehensive. Despite the main results, the authors also conduct a lot of analysis to verify the effectiveness of incorporating susceptibility risk.

In Figure 5, OSBP is a relatively old method (in 2018), whose performance is not outstanding in recent OSDA approaches. To a better understanding of the proposed framework, it is more appropriate to visualize the feature space of a more advanced method, especially ANNA (2023), which also explores the benefit of causality in OSDA.

**Methods And Evaluation Criteria:**

In the main paper, both Office31 and Image-CLEF utilized in experiments are quite simple benchmarks where the foreground objects and background in most of their samples are clearly distinguishable. I think the authors could conduct experiments on more challenging datasets (such as DomainNet), which contain more hidden causal relations that are suitable for evaluating the proposed framework.

With the above concerns, It is also suspectable for the application of this framework in practical scenarios. The proposed causal analysis seems restricted between single samples with clear foreground objects, while the open world often exists more complicated inter-object relations. Including an extended discussion for these situations will be better.

**Other Comments Or Suggestions:**

LLine 133-135, \epsilon in Equation 3 is not defined.

LLine 163-164, what does the abbreviation LB mean, couldn’t find it anywhere in the rest of this paper.

The reference to the anonymous code repository is unavailable.

**Other Strengths And Weaknesses:**

Refer to the above comments.

**Questions For Authors:**

Please refer to the above comments.

**Relation To Broader Scientific Literature:**

The proposed framework theoretically demonstrates the potential of causal inference in OSDA, which may inspire the following research to develop more robust and interpretable recognition systems that align with human thought in the open world.

**Theoretical Claims:**

The proofs in this paper seem correct without explicit logistic error.

---

> ### Author Rebuttal · Authors · 2025-04-01
>
> **Dear Reviewer rce5,**
>
> Thank you for your decision and constructive feedback. These detailed and professional comments have highly enlightened and encouraged us to make every effort to improve our work. We hope our responses could resolve the concerns.
>
> >**Claims And Evidence**.  What is the meaning of “c' is the specific implementation of C” in Definition 1.
>
> **A1**. We sincerely appreciate the opportunity to clarify this important conceptual point. We provide a simplified binary example to clarify the meaning. As shown in Fig. 7 (Appendix), for _maple leaf_ classification, let $C$ denote the presence of _palm-shaped lobes_. There the specific implementation $c\in \{0,1\}$:
> - $c=1$: Sample exhibits palm-shaped lobes
> - $c=0$: Absence of this trait
>
> Then the different implementations of $C$ in the sample will influnence the probablity of the label. We hope this illustrative example has clarified the causal feature representation.
>
> > **Methods And Evaluation Criteria**. I think the authors could conduct experiments on more challenging datasets (such as DomainNet)
>
> **A2**. We sincerely appreciate the suggestion. We have expanded our experiments on both DomainNet and VisDA.
>
> *Table 1. Comparison results on DomainNet and VisDA with CLIP backbone**( Baseline results from [R1]).***
> |Method|Domain|Net|(173/172)|VisDA|(6/6)||
> |-|-|-|-|-|-|-|
> ||OS*|UNK|HOS|OS*|UNK|HOS|
> |DCC[1]|50.2|45.1|47.5|75.3|46.2|57.3|
> |OVANet[2]|65.1|48.5|55.6|60.4|61.2|60.8|
> |CROW[R1]|70.3|50.9|59.0|77.0|62.8|69.2|
> |**COSDA-CLIP**|**72.0**|**76.2**|**73.9**|**85.2**|**72.6**|**78.4**|
>
> **[R1] Cross-domain Open-world Discovery, ICML 2024**
>
> **Key Findings from Table 1:**
> 1. Substantial HOS Improvements:
>     - On DomainNet, COSDA-CLIP achieves  **73.9% HOS, outperforming CROW by 14.9%**
>     - On VisDA, COSDA-CLIP reaches  **78.4% HOS, outperforming CROW by 9.2%**
> 2. Dual Strengths in OS\*  and UNK: With CLIP, COSDA particularly further enhances known-class classification and unknown-class detection (UNK).
>
> *Table 2.  Performance comparison with [1][3] on Office-Home and Office-31*
> |Method|Office-Home|Office-31|
> |-|-|-|
> |DCC[1]|64.2|86.8|
> |UADAL[3]|68.7|88.1|
> |**COSDA**|**71.7**|**92.6**|
>
> Table 2 shows leading performance of COSDA on small-scale datasets compared with [1][3]. These experiments could further confirm the effectiveness of COSDA.  **We will well cite [1][2][3] in the updated version.**
>
> >**Experimental Designs**: It is more appropriate to visualize the feature space of a more advanced method, especially ANNA (2023).
>
> **A3**. Thanks for your suggestions. We have added the requested comparison with ANNA in our feature space analysis. The visualizations can be found at https://anonymous.4open.science/r/tsne-5F2D/
>
> > **Essential References Not Discussed**: A close relative idea has been proposed in prior work DCC[1] published in CVPR 2021, which is not discussed in the related work. Several classical researches are missing, such as OVAnet [2] and UADAL [3].
>
> **A4**. We appreciate your insightful comments. **We will incorporate [1][2][3] into the updated related work.** DCC proposes Domain Consensus Clustering for universal domain adaptation, using semantical and sample-level consensus to effectively separate and distinguish common classes from private ones. OVANet proposes a universal domain adaptation method that learns an open-set threshold from source data via one-vs-all classifiers and adapts it to the target domain by minimizing class entropy. UADAL addresses open-set domain adaptation with unknown-aware adversarial learning, aligning known classes while segregating unknowns in feature space. We carefully read [1][2][3], and the difference is we exclude all known-class positive samples and cluster only the remaining negative samples (i.e., those not belonging to any known class). Besides, unlike one-vs-all strategies requiring C clustering operations per epoch (C = the number of known classes), COSDA achieves comparable performance with only one clustering per epoch.
>
> > Other Comments Or Suggestions
>
> **A5**. We sincerely appreciate your thorough review of our paper. **We will carefully correct the errors in the updated version and thoroughly review the text to prevent similar mistakes.**
>
> **LLine 133-135**: The parameter $\epsilon$ represents the degree of intervention on the causal features. It should be large to ensure that the semantic information can be disentangled (i.e., so that it influences the probability of label $Y$).
>
> **LLine 163-164**: The command `\LB` (which represents the label $Y$) was missing its backslash due to a typesetting error.
>
> **The anonymous code repository is unavailable**: We have updated the code in the previously provided anonymous link while also including necessary details for running it.
>
> We sincerely appreciate your insightful comments. Please let us know if you need any further information or if there are additional points you would like to discuss with us.
>
> Best regards,
>
> Authors of #10234

---

### Official Review · Reviewer_zv4D · 2025-03-10

**Overall Recommendation:** 3

**Summary:**

This paper introduces an adversarial adaptation framework called COSDA, which aims to address the challenges of unknown category recognition and domain drift in the open domain adaptation problem. The framework is based on causality theory and includes three novel components: (i) Susceptibility Risk Estimator (SRE), which is used to capture causal information and form a risk minimization framework; (ii) Contrastive Feature Alignment (CFA) module, which satisfies the external causal assumption and promotes cross-domain feature alignment based on mutual information theory proof; (iii) Virtual Multi-unknown-categories Prototype (VMP) pseudo-labeling strategy, which provides label information by measuring the similarity between samples and prototypes of known and multiple virtual unknown categories, thereby assisting open set recognition and intra-class discrimination learning. Experimental results show that the proposed method achieves state-of-the-art performance on benchmark datasets and synthetic datasets.

## update after rebuttal
Thank you very much for the author's reply. I maintain my initial positive rating.

**Claims And Evidence:**

Compared with the traditional OSDA method, the main improvement of CSDA is the introduction of causal inference technology and Susceptibility Risk Estimator (SRE), which enables the model to better handle open set problems and sources of uncertainty. In addition, CSDA also uses pseudo-labeling strategies and contrastive learning strategies for virtual multiple unknown categories to further improve the performance of the model.  This paper mainly introduces the causal inference based open domain adaptation method COSDA and conducts extensive experimental comparisons on three benchmark datasets.

**Essential References Not Discussed:**

No

**Experimental Designs Or Analyses:**

The experiments are reasonable.

**Methods And Evaluation Criteria:**

This paper conducts extensive experimental comparisons on three benchmark datasets. The experimental results show that COSDA achieves good performance on all benchmark datasets, especially when dealing with unknown categories. In addition, COSDA performs well for different evaluation indicators, such as unknown-class accuracy, known-class accuracy, overall accuracy, and harmonic mean accuracy. In the Ablation study, the authors further explored the impact of different components of COSDA on the performance. Finally, by applying COSDA to practical problems, the authors demonstrated its practical value in solving complex scenarios.

**Other Comments Or Suggestions:**

Please provide the code, which is important.

**Other Strengths And Weaknesses:**

strengths:

1) This paper proposes an OSDA method based on a causal model. By introducing the concept of probability sensitivity, a risk assessment framework for adversarial unknown categories is proposed, and a contrastive feature alignment and virtual multi-unknown category prototype strategy are designed to achieve open domain classification tasks.

2) Authors conduct expensive experiments, which show that this method achieves significant performance improvements on three benchmark datasets.

3) Paper is easy to follow. And the shown figures are interesting with enjoyable color and design.

Weaknesses:

1) As reported in the main tables, in most cases the proposed methods are not good and fail to be SOTA (State of the Art). This indicates that while the methods may show promise or have certain advantages in specific scenarios, they do not consistently outperform existing techniques across all metrics and benchmarks.

2) I think this task could potentially be completed by leveraging Video-Language Models (VLMs). However, the authors did not demonstrate scenarios or cases where VLMs might fail to work effectively. It's important for a comprehensive evaluation to include both the capabilities and limitations of such models.

3) No codes are provided, which is not too convincing. This lack of concrete examples makes it difficult to fully understand the implementation details and assess the validity of the claims being made. For a more robust evaluation, it's essential to have access to the specific code snippets or a complete codebase that demonstrates how the theoretical concepts are applied in practice. Without this, the explanation remains somewhat abstract and less actionable.

**Questions For Authors:**

No

**Relation To Broader Scientific Literature:**

This work could be also finished by VLM, which is not interesitng with tranditional method.

**Theoretical Claims:**

No

---

> ### Author Rebuttal · Authors · 2025-04-01
>
> **Dear Reviewer zv4D,**
>
> Thank you for your decision and constructive feedback. We have studied the comments carefully and made thorough revisions. We also greatly appreciate your insightful questions and hope that our responses have helped to clarify them.
>
> > **Weakness 1**: As reported in the main tables, in most cases the proposed methods are not good and fail to be SOTA (State of the Art). While the methods may show promise or have certain advantages in specific scenarios, they do not consistently outperform existing techniques across all metrics and benchmarks.
>
> **A1**.
> We appreciate the reviewer's attention to our experimental evaluation. Our comprehensive assessment includes three key metrics: **OS*** (known-class accuracy), **UNK** (unknown-class detection), and **HOS (balancing** both aspects). As highlighted in our contributions, COSDA achieves consistent improvements over SOTA methods:
> - **Absolute HOS gains**: +2.9% (Office-Home), +2.2% (Office-31), +1.0% (Image-CLEF)
> - **Dominant rankings**:
>   - Image-CLEF: Achieved the highest HOS in 11 out of 12 subtasks, and the best UNK score in 9 out of 12 subtasks.
>   - Office-31: Achieved the highest HOS in 3 out of 6 subtasks, and the best UNK score in 3 out of 6 subtasks.
>
> The method demonstrates both **generalizability** (highest average performance across all benchmarks) and **robustness** (most of the subtasks show statistically significant improvements). These results validate our design's effectiveness in handling the known-unknown class trade-off in many cases.
>
>
> > **Weakness 2**: I think this task could potentially be completed by leveraging Video-Language Models (VLMs). However, the authors did not demonstrate scenarios or cases where VLMs might fail to work effectively. It's important for a comprehensive evaluation to include both the capabilities and limitations of such models.
>
> **A2**. We sincerely appreciate this valuable suggestion.  We have conducted  additional experiments with  CLIP  ViT-L  on challenging datasets DomainNet and VisDA.
>
> *Table 1. Performance comparison (%) on DomainNet and VisDA datasets using CLIP backbone. **(All baseline results are obtained from [R1]).***
> |Method|Domain|Net|(173/172)|VisDA|(6/6)||
> |-|-|-|-|-|-|-|
> ||OS*|UNK|HOS|OS*|UNK|HOS|
> |DCC|50.2|45.1|47.5|75.3|46.2|57.3|
> |UNIOT|59.2|45.1|51.2|75.7|49.4|59.8|
> |GLC|62.9|50.6|56.1|73.4|58.7|65.2|
> |CROW[R1]|70.3|50.9|59.0|77.0|62.8|69.2|
> |COSDA-CLIP|**72.0**|**76.2**|**73.9**|**85.2**|**72.6**|**78.4**|
>
> **[R1]Cross-domain Open-world Discovery, ICML 2024**
>
> **Implementation Details**.  Considering both time constraints and GPU memory demands (particularly for the larger models), we utilized six 40GB NVIDIA A100 GPUs to execute the new experiments. DomainNet and VisDA use the same hyperparameter settings as smaller-scale datasets, specifically $\lambda_s=0.2$, $\lambda_{exo}=1$. But the learning rate has been reduced,  specifically $lr = 5e-4$.
>
> **Key Findings from CLIP Backbone Experiments:**
> 1.  _Substantial HOS Improvements_:
>     -   On DomainNet, COSDA-CLIP achieves  **73.9% HOS, outperforming CROW by 14.9%**.
>     -   On VisDA, COSDA-CLIP reaches  **78.4% HOS, outperforming CROW by 9.2%** .
> 2.  _Dual Strengths in OS\*  and UNK_: With CLIP, COSDA particularly further enhances known-class classification and unknown-class detection (UNK).
>     - For DomainNet OS*, a 1.7% improvement;
>     - For DomainNet UNK, a 25.3% improvement;
>     - For VisDA OS*, an 8.2% improvement;
>     - For VisDA UNK, a 9.8% improvement.
>
> *Table 2. Sub-Task Performance of COSDA on DomainNet with CLIP bakcbone*
> |subtask|P-R|P-S|R-P|R-S|S-P|S-R|**Avg.**|
> |-|-|-|-|-|-|-|-|
> |OS\*|76.6|68.8|71.7|70.9|66.1|78.1|72.0|
> |UNK|78.9|79.1|72.2|77.2|76.3|73.2|76.2|
> |HOS|77.7|73.6|71.9|73.9|70.8|75.6|73.9|
>
> Table 2 exhibited detailed results of COSDA for 6 subtasks on DomainNet with CLIP. These additional experiments could further confirm the effectiveness of COSDA. **We fully agree with the reviewer's insightful suggestion about VLMs. This represents a promising new direction worth exploring in OSDA.**
>
> > **Weakness 3 & Other Comments**: No codes are provided.
>
> **A3**. We sincerely appreciate the reviewer's emphasis on reproducibility. **As noted in our submission (Page 6, Line 321), we open-sourced the complete implementation. For greater visibility, we will relocate the code announcement to the abstract/introduction in the updated version.**
>
> **Current Implementation Overview:**
> - **Benchmark Support**: Office-Home, Office-31, Image-CLEF, DomainNet, VisDA
> - **Architecture Flexibility**:
>   - CNN backbones (ResNet, VGG)
>   - VLMs (CLIP)
> - **Training Frameworks**:
>   - Multi-GPU distributed training
>   - Single-GPU training
>
> We sincerely appreciate your insightful comments once again. Please let us know if you need any further information or if there are additional points you would like to discuss with us.
>
> Best regards,
>
> Authors of #10234

---

### Official Review · Reviewer_pQYc · 2025-03-12

**Overall Recommendation:** 3

**Summary:**

This paper addresses the Open-Set Domain Adaptation problem which is useful in real-world applications. They propose a novel Counterfactual-based susceptibility risk framework, consists of Susceptibility Risk Estimator, Contrastive Feature Alignment, and Virtual Multi-unknown-categories Prototype. Experiments on three datasets and benchmarks highlight its superior performance.

**Claims And Evidence:**

Yes

**Essential References Not Discussed:**

No

**Experimental Designs Or Analyses:**

The experimental designs are sound in general. However, the experiments on large-scale benchmark such as DomainNet and VisDA are missing.

**Methods And Evaluation Criteria:**

The proposed method follows the traditional evaluation criteria in Open-Set domain adaptation.

**Other Comments Or Suggestions:**

There are too many loss terms in eq (25) which is complex and may make optimization difficult. How do you balance the contribution of different terms?

**Other Strengths And Weaknesses:**

Strengths
1. The paper addresses the Open-Set domain adaptation problem, which is a challenging and practical scenario.
2. The paper is well written and easy to follow.
3. The paper provides extensive experiments, showing the effectiveness and versatility of the proposed method.

Major Weaknesses
1. The authors only use CNN backbones. More ablations on ViT backbone should be added, as it demonstrates strong generalization and adaptation performances compared with CNNs.
2. Although it is important for Open-Set domain adaptation to have a good performance on UNK, it is also crucial to have a good OS* value. However, the OS* of the proposed framework is worse than baselines on all datasets. For example, 0.3 on Image-CLEF, 3.2 on Office-31, and 6.7 on Office-Home.
3. Lack of experiments on large-scale benchmarks such as DomainNet and VisDA, which are commonly used in existing work [1,2].

[1] Upcycling Models under Domain and Category Shift, CVPR 2023

[2] LEAD: Learning Decomposition for Source-free Universal Domain Adaptation, CVPR 2024

**Questions For Authors:**

What's the performances of the proposed framework on DomainNet and VisDA?

**Relation To Broader Scientific Literature:**

The proposed Counterfactual-based susceptibility risk framework could be potentially helpful to other literature.

**Theoretical Claims:**

Yes, no issues.

---

> ### Author Rebuttal · Authors · 2025-04-01
>
> **Dear Reviewer pQYc,**
>
> We sincerely appreciate your constructive comments on our work. We have carefully addressed each point raised and incorporated corresponding improvements. We hope our responses have adequately addressed the concerns.
>
> >**Weaknesses 1 & Weaknesses 3**.  More ablations on ViT backbone should be added; Lack of experiments on large-scale benchmarks such as DomainNet and VisDA.
>
> **A1**.  We fully agree with the reviewer's valuable suggestion regarding the importance of evaluating our method on large-scale datasets with diverse backbones, as this would further validate the generalization of COSDA. In response to this suggestion, we implemented additional experiments on VisDA and DomainNet [1][2][R1][R2] and used CLIP ViT-L14-336px as the backbone.
>
> **[R1]Cross-domain Open-world Discovery, ICML 2024
> [R2] Domain consensus clustering for universal domain adaptation, CVPR 2021**
>
> **Implementation Details**. Considering both time constraints and GPU memory demands, we utilized six 40G A100 GPUs to execute the new experiments. DomainNet and VisDA use the same hyperparameter settings as smaller-scale datasets, specifically $\lambda_s=0.2$ and $\lambda_{exo}=1$. The learning rate is $5e-4$.
>
> *Table 1. Performance comparison (%) on DomainNet and VisDA using CLIP backbone. **(Baseline results from [R1]).***
> |Method|Domain|Net|(173/172)|VisDA|(6/6)||
> |-|-|-|-|-|-|-|
> ||OS*|UNK|HOS|OS*|UNK|HOS|
> |DCC|50.2|45.1|47.5|75.3|46.2|57.3|
> |UNIOT|59.2|45.1|51.2|75.7|49.4|59.8|
> |GLC[1]|62.9|50.6|56.1|73.4|58.7|65.2|
> |CROW[R1]|70.3|50.9|59.0|77.0|62.8|69.2|
> |COSDA-CLIP|**72.0**|**76.2**|**73.9**|**85.2**|**72.6**|**78.4**|
>
> **Key Findings from CLIP Backbone Experiments:**
> 1. Substantial HOS Improvements:
>     - On DomainNet, COSDA-CLIP achieves  **73.9% HOS, outperforming CROW by 14.9%**.
>     - On VisDA, COSDA-CLIP reaches  **78.4% HOS, outperforming CROW by 9.2%** .
> 2. Dual Strengths in OS\*  and UNK: With CLIP, COSDA particularly further enhances known-class classification and unknown-class detection (UNK).
>     - For DomainNet OS*, a 1.7% improvement;
>     - For DomainNet UNK, a 25.3% improvement;
>     - For VisDA OS*, an 8.2% improvement;
>     - For VisDA UNK, a 9.8% improvement.
>
> *Table 2. Performance comparison on VisDA (VGG19). **(Baseline results from [R2]).***
> |Metric|OSBP|STA|DCC[R2]|COSDA|
> |-|-|-|-|-|
> |OS*|62.9|66.8|68.8|**80.7**|
> |OS|59.2|63.9|68.0|**70.1**|
>
> *Table 3. Sub-Task Performance of COSDA Across Backbones on DomainNet*
> |Subtask|CLIP|||ResNet|50||
> |-|-|-|-|-|-|-|
> ||OS*|UNK|HOS|OS*|UNK|HOS|
> |P-R|76.6|78.9|77.7|65.4|43.4|52.2|
> |P-S|68.8|79.1|73.6|55.1|76.9|64.2|
> |R-P|71.7|72.2|71.9|45.1|56.6|50.2|
> |R-S|70.9|77.2|73.9|42.4|49.0|45.5|
> |S-P|66.1|76.3|70.8|50.8|79.2|61.9|
> |S-R|78.1|73.2| 75.6|70.8|85.9|77.6|
> |**Avg.**|72.0|76.2|73.9|54.9|65.2|58.6|
>
> *Table 4.  Performance comparison with [1][2] on Office-Home and Office-31*
> |Method|Office-Home|Office-31|
> |-|-|-|
> |GLC[1]|69.8|89.0|
> |LEAD[2]|70.0|90.1|
> |**COSDA**|**71.7**|**92.6**|
>
> Table 2 shows leading performance on VisDA (VGG-19). Table 3 exhibited detailed results of COSDA for 6 subtasks on DomainNet with ResNet-50 and CLIP. Table 4 supplements the performance comparison between COSDA and references [1][2] on small-scale datasets. These additional experiments could further confirm the effectiveness of COSDA. **Additionally, we will well cite [1][2] you provided in the updated revisions**.
>
> > **Weaknesses 2**. The OS* of the proposed framework is worse than baselines on all datasets. For example, 0.3 on Image-CLEF, 3.2 on Office-31, and 6.7 on Office-Home.
>
> **A2**. We sincerely appreciate the reviewer's insightful observation regarding the OS* performance. We acknowledge that our method demonstrates a modest compromise in OS* scores. However, this trade-off enables significant improvements in UNK (33.2%, 43.4%, and 49.4% gains, respectively). HOS is the most crucial metric, which achieves a balance between known-class and unknown-class recognition. For HOS, COSDA outperforms the highest OS* methods by 19.3%, 28.1%, and 36.5% on three small-scale datasets. Notably, **COSDA achieves OS\*, UNK, and HOS improvement on challenging DomainNet (Table 1)**. These results could substantiate the advantages of our approach.
>
> > **Other Comments**. There are too many loss terms in Eq (25) which is complex and may make optimization difficult. How do you balance the contribution of different terms?
>
> **A3**. Thank you for raising this important point. To balance the contributions of different loss terms in the overall objective, we introduced weight hyperparameters for loss items. These hyperparameters were optimized via a grid search strategy to ensure an appropriate trade-off for the dataset. As shown in Fig. 6, we analyzed the impact of parameters.
>
> We sincerely appreciate your insightful comments again. Please let us know if you need any further information or if there are additional points you would like to discuss with us.
>
> Best regards,
>
> Authors of #10234

---

> > ### Comment · Reviewer_pQYc · 2025-04-07
> >
> > I want to thank the authors for the rebuttal, most of my concerns are addressed and I therefore increase the score to 3.

---

> > > ### Author Response · Authors · 2025-04-07
> > >
> > > Dear Reviewer pQYc,
> > >
> > > We sincerely appreciate your great support, which means a great deal to us! Engaging in this discussion with you has been truly rewarding.
> > >
> > > Thank you once again for your valuable time and effort !
> > >
> > > Best regards,
> > >
> > > Authors of #10234

---

### Official Review · Reviewer_6vAQ · 2025-03-13

**Overall Recommendation:** 4

**Summary:**

This paper introduces COSDA, a novel causal-based Open-Set Domain Adaptation (OSDA) framework. It proposes Susceptibility Risk, a theoretical approach to measuring and mitigating the risk associated with domain shifts and unknown category recognition. Then, three core components are developed: Susceptibility Risk Estimater (SRE), Contrastive Feature Alignment (CFA), and Virtual Multi-unknown-categories Prototype (VMP), all of which contribute to better feature alignment and improved classification of unknown categories. Extensive experiments on benchmark datasets (Office-Home, Office-31, and Image-CLEF) demonstrate that COSDA outperforms state-of-the-art methods, achieving significant improvements in accuracy and robustness.

## update after rebuttal

**Claims And Evidence:**

The claims are well supported by the evidence.

**Essential References Not Discussed:**

-

**Experimental Designs Or Analyses:**

In general, the soundness of experimental designs and analyses is strong. However, since the experiments were conducted on Image-CLEF, Office-31, and Office-Home datasets, whether the method can perform well on harder data (like DomainNet [1] QuickDraw) remains unknown.

[1] Moment matching for multi-source domain adaptation, ICCV 2019

**Methods And Evaluation Criteria:**

The method and evaluation criteria make sense for the problem.

**Other Comments Or Suggestions:**

-

**Other Strengths And Weaknesses:**

-

**Questions For Authors:**

1. ImageNet-pretrained ResNet 50 is used for all the experiments. However, as discussed in [1] and [2], all the conclusions and insights can change when using a foundation model like CLIP or DINO_v2 as the backbone. Also, given that stronger pretrained backbones generally enhance image classification performance, it is unclear why ResNet-50 remains the preferred choice. Could you discuss this decision and its potential impact on the findings?
2. The step of building the centroids of all known and unknown classes shares the same idea as in [2] and [3]. Could you please discuss the main difference between method level and motivation level?
3. The motivation for using causality is strong and clear. However, the motivation behind the overall method design is difficult to follow.. The framework consists of multiple components (SRE, CFA, and VMP), but their individual necessity and how they collectively contribute to the causal inference remain unclear. Could you clarify the motivation behind designing these three components, their specific roles in causal inference, and how they interconnect within the overall methodology?

[1] Universal domain adaptation from foundation models: A baseline study

[2] Cross-domain Open-world Discovery, ICML 2024

[3] Domain consensus clustering for universal domain adaptation, CVPR 2021

**Relation To Broader Scientific Literature:**

This work views the OSDA setting from the causal inference perspective and bridges the gap between causal-inspired theoretical frameworks and OSDA.

**Theoretical Claims:**

There is no obvious issue in all the definitions, lemmas, propositions, theorems, and corollaries.

---

> ### Author Rebuttal · Authors · 2025-03-31
>
> **Dear Reviewer 6vAQ,**
>
> Thank you for your decision and constructive feedback. We have stuied the comments carefully and made through revisions. We hope that our responses have helped to clarify the concerns.
> > **Experimental Designs Or Analyses & Q1**. whether the method can perform well on harder data (like DomainNet [1]) remains unknown ;  All the conclusions and insights can change when using different backbones. Why ResNet-50 remains the preferred choice?
>
> **A1**. We sincerely appreciate your insightful suggestions and questions. Initially, we adopted ResNet50 as the backbone to maintain consistency with prior work in this domain. However, as you rightly pointed out, evaluating the model with more advanced backbones and larger-scale datasets would better demonstrate its robustness. Following this suggestion, we implemented addtional experiments on VisDA[3] and DomainNet[1][2] and discovered that **our method demonstrates good performance on both CNN-based and CLIP-based architectures**.
>
> *Table 1. Comparison results on DomainNet and VisDA with CLIP backbone **(All baseline results are obtained from Reference [2]).***
> |Method|Domain|Net|(173/172)|VisDA|(6/6)||
> |-|-|-|-|-|-|-|
> ||OS*|UNK|HOS|OS*|UNK|HOS|
> |DCC[3]|50.2|45.1|47.5|75.3|46.2|57.3|
> |UNIOT[1]|59.2|45.1|51.2|75.7|49.4|59.8|
> |CROW[2]|70.3|50.9|59.0|77.0|62.8|69.2|
> |**COSDA-CLIP**|**72.0**|**76.2**|**73.9**|**85.2**|**72.6**|**78.4**|
>
> **Implementation Details**.  Considering both time constraints and GPU memory demands (particularly for the larger models), we utilized six 40GB NVIDIA A100 GPUs to execute the new experiments. DomainNet and VisDA use the same hyperparameter settings as smaller-scale datasets, specifically $\lambda_s=0.2$, $\lambda_{exo}=1$. But the learning rate has been reduced,  specifically $lr = 5e-4$.
>
> **Key Findings from CLIP Backbone Experiments:**
> 1. Substantial HOS Improvements:
>     - On DomainNet, COSDA-CLIP achieves  **73.9% HOS, outperforming CROW by 14.9%**.
>     - On VisDA, COSDA-CLIP reaches  **78.4% HOS, outperforming CROW by 9.2%** .
> 2. Dual Strengths in OS\*  and UNK: With CLIP, COSDA particularly further enhances known-class classification and unknown-class detection (UNK).
>
> *Table 2. Performance of COSDA on DomainNet with ResNet50*
> ||ResNet50|||
> |-|-|-|-|
> ||OS\*|UNK|HOS|
> |**Avg.**|54.9|65.2|58.6|
>
> We also note that with ResNet50, COSDA surpasses DCC [3] and UNIOT [1], confirming its robustness across backbones.  **The suggested references will be well cited in our updated version.**
>
> > **Q2**. The step of building the centroids of all known and unknown classes shares the same idea as in [2] and [3]. Please discuss the main difference between method level and motivation level.
>
> **A2**. We sincerely appreciate your insightful questions. In our work, the **VMP** module serves as a functional component to ensure **CFA**, rather than acting as a standalone clustering solution. I carefully read the papers you provided, and the difference lies in the fact that we **exclude all known-class positive samples and cluster only the remaining negative samples** (i.e., those not belonging to any known class), unlike [2][3], which cluster _all target_ samples. Due to the reduction in the number of samples and clusters, VMP has saved on the computational cost. Besides, unlike one-vs-all strategies requiring C clustering operations per epoch (C = the number of known classes), COSDA achieves comparable performance with only one clustering per epoch. This design reduces complexity from O(CN) to O(N). Additionally, we acknowledge that advanced clustering could further optimize this step and will explore this in future work.
>
> > **Q3**. Could you clarify the motivation behind designing SRE, CFA, and VMP, their specific roles in causal inference, and how they interconnect within the overall methodology?
>
> **A3**. We appreciate your questions regarding the methodological framework. SRE quantifies causal feature representation ability via susceptibility analysis. Direct SRE estimation in the target domain is infeasible due to label scarcity. To resolve this problem, we first decompose target-domain expected risk into open-set and closed-set, then bridge source/target risks using domain-invariant representations, and finally derive generalization bounds to narrow the gap between empirical and expected susceptibility risks (Theorems 1-2 & Corollary 1). To satisfy the exogeneity assumption for causal identifiability, we propose CFA, which encourages independence across causal features belonging to different categories via information bottleneck (Proposition 2) and then introduces VMP pseudo-label strategy (Eq. 17-19). Additionally, our code is open-sourced to show how SRE, CFA, and VMP are integrated end-to-end.
>
> We sincerely appreciate your insightful comments once again. Please let us know if you need any further information or if there are additional points you would like to discuss with us.
>
> Best regards,
>
> Authors of #10234

---

> > ### Comment · Reviewer_6vAQ · 2025-04-02
> >
> > Thank you for your effort during the rebuttal. I am glad to see your method working well with the CLIP backbone and outperforming CROW, the current state-of-the-art, by a large margin. Also, the explanations of Q2 and Q3 are clear. I will update the score from 3 to 4.
> >
> > Some suggestions to update the current draft:
> >
> > 1. The experiments in Q1 can be added to a new section 4.6, showing the robustness across different backbones, especially the strong backbones (foundation model).
> >
> > 2. The answer to Q3 clearly explains the motivation for the design of the method. It can be added at the beginning or end of the method part. Right after section 3.1 or at the end of section 3 as a summary. Since this approach consists of many parts, it is better to provide the reader with a clearer and logical understanding of the design.

---

> > > ### Author Response · Authors · 2025-04-03
> > >
> > > Dear Reviewer 6vAQ,
> > >
> > > We sincerely appreciate your great support, which means a great deal to us! In response to your suggestions, we will add the experiments in Q1 to a new section 4.6, and add our response to Q3 at the beginning or end of the Methods section.
> > >
> > > Once again, we are deeply grateful for your support and guidance.
> > >
> > > Best regards,
> > >
> > > Authors of #10234

---

### Decision · Program_Chairs · 2025-05-01

**Decision:**

Accept (poster)

**Comment:**

Reviewers appreciated the paper including both the theoretic and methodological aspects. While reviewers noted a few weaknesses, the authors' rebuttal addressed the main concerns.

Please make sure to address the comments from the reviewers in the final version to improve the paper.